# Single-cell analysis of peripheral blood from high-altitude pulmonary hypertension patients identifies a distinct monocyte phenotype

Xin-Hua Wu [1,11], Yang-Yang He [2,11], Zhang-Rong Chen [1,3,11], Ze-Yuan He[4], Yi Yan [5], Yangzhige He[6], Guang-Ming Wang[1], Yu Dong[1], Ying Yang[1], Yi-Min Sun[7], Yong-Hong Ren[7], Qiu-Yan Zhao[1], Xiao-Dan Yang[1], Li-Ying Wang[1], Cai-Jun Fu[1], Miao He[8], Si-Jin Zhang[9], Ji-Fen Fu[1], Hong Liu [1] ✉ & Zhi-Cheng Jing [10] ✉

Immune and inflammatory responses have an important function in the pathophysiology of pulmonary hypertension (PH). However, little is known about the immune landscape in peripheral circulation in patients with high-altitude pulmonary hypertension (HAPH). We apply single-cell transcriptomics to characterize the monocytes that are significantly enriched in the peripheral blood mononuclear cells (PBMC) of HAPH patients. We discover an increase in C1 (non-classical) and C2 (intermediate) monocytes in PBMCs and a decrease in hypoxia-inducible transcription factor-1α (HIF-1α) in all monocyte subsets associated with HAPH. In addition, we demonstrate that similar immune adaptations may exist in HAPH and PH. Overall, we characterize an immune cell atlas of the peripheral blood in HAPH patients. Our data provide evidence that specific monocyte subsets and HIF-1α downregulation might be implicated in the pathogenesis of HAPH.

Pulmonary hypertension (PH) encompasses a group of severe clinical disorders that were originally defined by a resting mean pulmonary arterial pressure (mPAP) of 20 mm Hg or above[1]. An updated clinical classification of PH was established, in which the World Health Organization (WHO) categorized PH into five groups of disorders with similar pathological and hemodynamic characteristics: group 1—pulmonary arterial hypertension (PAH), group 2—PH due to left heart disease, group 3—PH due to lung diseases and/or hypoxia; group 4—chronic thromboembolic PH and other pulmonary artery obstructions, and group 5—PH with unclear and/or multifactorial mechanisms[2].

High-altitude pulmonary hypertension (HAPH) belongs to the third category of PH, which affects populations residing at altitudes above 2500 meters. HAPH has become a public health problem in the high-altitude areas of the world, including the Kyrgyz highlands,

[1]Department of Cardiology; Yunnan Provincial Engineering Research Center of Prevention and Treatment of Trans-plateau Cardiovascular Diseases, The First Affiliated Hospital of Dali University, Yunnan, China. [2]School of Pharmacy, Henan University, Henan, China. [3]Department of Cardiology, The Affiliated Hospital of Guizhou Medical University, Guizhou, China. [4]Department of Cardiology, Yulong People's Hospital, Yunnan, China. [5]Heart Center and Shanghai Institute of Pediatric Congenital Heart Disease, Shanghai Children's Medical Center, National Children's Medical Center, Shanghai Jiao Tong University School of Medicine, Shanghai, China. [6]Department of Medical Research Center, State Key Laboratory of Complex Severe and Rare Diseases, Peking Union Medical College Hospital, Chinese Academy of Medical Sciences & Peking Union Medical College, Beijing, China. [7]CapitalBio Technology Corporation, Beijing, China. [8]Institute of Pharmacy, Dali University, Yunnan, China. [9]Department of Cardiology, Ruijin Hospital Affiliated to Shanghai Jiao Tong University School of Medicine, Shanghai, China. [10]Department of Cardiology, State Key Laboratory of Complex Severe and Rare Diseases, Peking Union Medical College Hospital, Chinese Academy of Medical Sciences & Peking Union Medical College, Beijing, China. [11]These authors contributed equally: Xin-Hua Wu, Yang-Yang He, Zhang-Rong Chen. ✉e-mail: daliliuhong@163.com; jingzhicheng@vip.163.com

Ethiopia, Andean regions, and the Qinghai–Tibet Plateau of China[3, 4]. Despite the availability of multiple treatment strategies targeting the endothelin, nitric oxide and prostacyclin pathways, these medications can only help relieve the symptoms and slow the progress of PH[3, 5, 6].

It is increasingly acknowledged that PH is a progressive vascular disorder due to a variety of pathological processes, such as vascular cell accumulation, endothelial dysfunction, smooth muscle cell proliferation, epigenetic alteration, metabolic reprogramming and perivascular inflammation[7–9]. However, the mechanisms underpinning HAPH are less well understood. A previous investigation showed that the pathogenic factors contributing to HAPH progression may include hypoxia, vasculopathy, nitric oxide (NO) synthesis, and metabolic abnormalities[6, 10]. Specifically, the hypoxic stimuli of high altitudes can trigger pulmonary vasoconstriction and increase pulmonary arterial pressure (PAP)[6], and hypoxia-inducible transcription factors (HIFs) is involved in the pathophysiology of pulmonary vascular remodeling[11–13]. Additionally, a recent experimental study showed that nonclassical monocytes in mouse lung tissues could sense hypoxia, infiltrate small pulmonary arteries, and promote PH[14]. Since there is not a reliable method for obtaining lung tissues in PH patients, investigations of the mechanisms contributing to disease progression have been hampered. Recently, high-throughput functional genomic or proteomic approaches have been applied to peripheral blood samples to explore PH pathobiology[15, 16]. Meta-analyses of blood expression profiles in PH suggested shared immunologic changes with their comparable histopathologies[15, 17]. Collectively, these studies indicated that a sustained dysregulated immune response may contribute to the initiation or progression of PH. However, until now, little has been known about the whole genome-wide expression profiling of peripheral blood mononuclear cells (PBMCs) and the circulating immune response of HAPH patients.

Here, we show that HAPH patients display a significant increase in the frequency of specific monocyte subsets in PBMCs compared to controls. Moreover, HAPH patients may suffer from impairment of HIF-1α signaling in the monocytes of the blood, displaying a distinct gene signature from that of controls. Collectively, we depict a cell atlas of the circulating immune response in HAPH patients and highlight a potential link between peripheral monocytes and the pathophysiology of HAPH.

## Results

### Single-cell RNA profiling of PBMCs in HAPH patients and healthy controls

6 PH patients living at 45 m altitude, 11 HAPH patients and 15 contemporary individuals living at 2900–3100 m altitude were enrolled in this study (Fig. 1). There were no differences between the three groups in blood pressure, heart rate, cholesterol, or hepatic and renal function. The sPAP was higher in PH patients compared to that of HAPH patients ($86.5 \pm 16.0$ mmHg vs. $50.0 \pm 9.8$ mmHg, $p < 0.05$) and both groups had higher sPAP than controls with sPAP averaged at $10.9 \pm 4.4$ mmHg (both $p < 0.05$) (Table 1). To analyze the circulating immunological changes associated with HAPH, we performed scRNA-seq on immune cells isolated from 7 HAPH patients and 5 healthy controls (Fig. 1). We assessed an average of 5733 cells per sample, with a median of 1204 genes detected per cell.

### Differential immune cell compositions in PBMCs between HAPH patients and healthy controls

To characterize the immune landscape and its association with HAPH, unsupervised clustering of the cells was performed (Supplementary Fig. 1 and 2). In all samples, we identified 9 major cell types consisting of T cells (e.g., CD4+ T cells, CD8+ T cells, and regulatory T cells), B cells (e.g., B cells, Memory B cells, and Naïve B cells), monocytes, NK cells, and hematopoietic stem cells (HSCs) according

to the canonical gene marker expressions (Fig. 2a, b). We could also identify a few dendritic cells with the classical marker gene CD1C (Supplementary Fig. 3). When comparing the cluster compositions between the two groups, we observed that cluster 16 (monocytes) was enriched in the HAPH patients ($p < 0.05$, two-sided Wilcoxon rank-sum test), and clusters 18 (NK cells) and cluster 21 (CD4+ T cells) were more frequent in the control samples ($p < 0.05$ for all comparisons, two-sided Wilcoxon rank-sum test) (Supplementary Fig. 2a, b). Next, we analyzed the relative percentages of the above main cell types in both groups. It turned out that only monocytes were significantly enriched in the HAPH patients ($p < 0.05$, two-sided Wilcoxon rank-sum test) (Fig. 2c, d). The relative abundances of NK cells tended to be enriched in the control samples, although the differences did not reach statistical significance (Fig. 2c, d). To better understand the nature of the immune profiles, we validated T cells, and NK cells by flow cytometry with classical markers for the cell gating. The frequency of immune subpopulations in PBMCs were shown in Supplementary Fig. 2c (Supplementary Table 1). These observations suggest that a dysregulated inflammatory response may occur in patients with HAPH.

### Defining monocyte subsets and their associations with HAPH

Human monocytes consist of phenotypically and functionally distinct subpopulations[18]. Previous studies have indicated a critical contribution of monocytes to the pathophysiology of PH[14, 19, 20]. Based on the high frequency of monocytes and their association with HAPH, we therefore focused our analysis on monocytes. Unsupervised clustering was performed on all monocytes, which identified three major subclusters: monocytes C0, C1 and C2 (Fig. 3a–c). The differentially expressed gene lists were shown in Supplementary Table 2 for C0, C1 and C2, respectively. Monocyte C0 cells expressed the classical monocyte markers of CD14, phospholipase B domain containing 1 (PLBD1), pro-inflammatory mediators S100 calcium binding protein A2

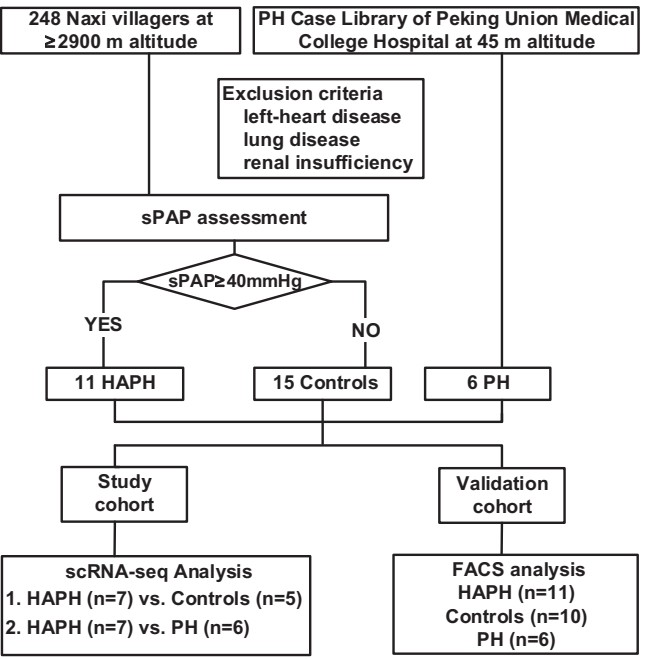

**Fig. 1 | Flowchart of study design.** In total, 11 patients with sPAP ≥40 mmHg and 15 controls who lived at the ≥2900 m altitude, 6 PH cases living at 45 m altitude were enrolled. scRNA-seq analysis was performed comparing between 7 HAPH and 5 controls, and also between 7 HAPH and 6 PH, followed by validation carried out among 6 PH, 11 HAPH and 10 controls with FACS analysis. PH pulmonary hypertension, HAPH high-altitude pulmonary hypertension, sPAP systolic pulmonary artery pressure, scRNA-seq single-cell RNA sequencing, FACS fluorescence-activated cell sorting.

**Table 1 | Characteristics of the participants in the study**

| Variables | PH (*n* = 6) | HAPH (*n* = 11) | Control (*n* = 15) |
|---|---|---|---|
| Gender (male/female, n) | 2/4 | 2/9 | 5/10 |
| Age (years) | 60.8 ± 9.9 | 60.6 ± 11.1 | 55.5 ± 10.8 |
| Systolic blood pressure (mmHg) | 123.8 ± 13.9 | 145.0 ± 13.1 | 143.5 ± 17.0 |
| Diastolic blood pressure (mmHg) | 84.2 ± 10.0 | 77.7 ± 14.2 | 81.5 ± 10.5 |
| Pulse blood oxygen saturation (%) | 0.89 ± 0.11 | 0.94 ± 0.04 | 0.94 ± 0.03 |
| Body mass index (kg/m$^2$) | 22.86 ± 4.14 | 23.91 ± 3.90 | 23.06 ± 2.66 |
| **Blood routine** | | | |
| White blood cell count (×10$^9$/L) | 7.0 ± 2.3 | 5.8 ± 2.1 | 6.2 ± 1.3 |
| Red blood cell count (×10$^{12}$/L) | 5.0 ± 1.2 | 4.9 ± 0.3 | 2.2 ± 0.3 |
| Hemoglobin (g/L) | 161.8 ± 38.9 | 149.8 ± 17.8 | 158.1 ± 15.7 |
| Platelet count (×10$^9$/L) | 187.2 ± 62.7 | 239.6 ± 65.7 | 233.4 ± 72.3 |
| **Blood chemistry** | | | |
| Aspartate amino transferase (U/L) | 23.7 ± 6.1 | 41.4 ± 27.8 | 23.9 ± 2.7 |
| Total protein (g/L) | 65.0 ± 15.0 | 76.0 ± 4.5 | 76.3 ± 4.5 |
| Albumin (g/L) | 39.5 ± 4.1 | 42.5 ± 4.4 | 44.3 ± 2.5 |
| Total bilirubin (μmol/L) | 19.5 ± 8.9 | 17.8 ± 10.9 | 15.3 ± 7.3 |
| Uric acid (μmol/L) | 409.4 ± 60.2 | 342.9 ± 131.2 | 327.8 ± 129.5 |
| Creatinine (μmol/L) | 92.3 ± 44.4 | 58.3 ± 19.9 | 64.5 ± 18.1 |
| Fasting blood glucose (mmol/L) | 5.6 ± 2.4 | 4.6 ± 0.3 | 5.6 ± 1.3 |
| **Echocardiography** | | | |
| Pulmonary Artery Dimension (mm) | 33.7 ± 8.7 | 20.1 ± 2.0 | 20.7 ± 3.7 |
| Left ventricular ejection fraction (%) | 75.2 ± 9.1 | 64.4 ± 8.6 | 65.3 ± 11.8 |
| Right atrium diameter (mm) | 56.5 ± 4.8 | 41.7 ± 2.5 | 38.4 ± 3.7 |
| Right ventricle diameter (mm) | 39.2 ± 5.0 | 27.2 ± 8.1 | 26.0 ± 7.1 |
| TVRVmax (m/s) | 4.5 ± 0.5 | 3.3 ± 0.3 | 1.3 ± 0.5 |
| Pulmonary systolic pressure (mmHg) | 86.5 ± 16.0 | 50.0 ± 9.8 | 10.9 ± 4.4 |

(S100A12) and S100A8, while the Fc fragment of the IgG receptor IIIa (FCGR3A)/CD16 was rarely expressed (Fig. 3d), which was similar to the major population of monocytes, classical (CD14$^+$CD16$^-$) monocytes[18, 21, 22]. Monocyte C1 cells expressed the highest level of FCGR3A/CD16 and median levels of CD14, with upregulation of the ras homolog gene family member C (RHOC), heme oxygenase 1 (HMOX1), Src family kinases constituting hemopoietic cell kinase (HCK) and tyrosine protein kinase Lyn (LYN) and complement component 1 q subcomponent-α polypeptide (C1QA) (Fig. 3d). The RNA expression profiles of monocyte C1 cells were similar to those of non-classical (e.g., CD14$^-$CD16$^+$) monocytes, as reported previously[22, 23]. While monocyte C2 cells exhibited relatively lower expressions of FCGR3A/CD16 (Fig. 3d), which were similar to those of intermediate (e.g., CD14$^-$CD16$^-$) monocytes[21, 23].

Next, we analyzed the frequencies of the three monocyte subpopulations between the case and control groups. Intriguingly, we determined that both C1 and C2 monocytes were more frequent in the patients (Fig. 3e, f, *p* < 0.05 for all comparisons, two-sided Wilcoxon rank-sum test). Specifically, the C1 (non-classical) and C2 (intermediate) monocytes accounted for 1.35% ~2.01% and 0.35% ~0.67%, respectively, of the total cells analyzed in PBMCs from the control individuals, while the frequencies increased to 1.71% ~5.26% and 1.09%

~9.62%, respectively, in HAPH patients (Fig. 3e, f). Significant higher frequencies of C1 and C2 were displayed in HAPH patients relative to that of controls (Fig. 3g–i). The correlation coefficient of clinical parameters and fractions of monocytes subtypes were shown in Supplementary Table 3. Thus, our data indicated that C1 (non-classical) and C2 (intermediate) monocytes may be instrumental in the pathophysiology of HAPH.

**Functional characterization of different monocyte subsets associated with HAPH**

To better understand the global functions of different monocyte subsets in HAPH development, we explored the DEGs and pathways in each monocyte subset between cases and controls. As expected, a large number of DEGs were identified in each monocyte subset (see Supplementary Data 1–3), and the top 100 DEGs were enriched in various pathways, including responses to virus, regulation of lymphocyte activation, leukocyte cell-cell adhesion, interleukin-2 (IL-2) production, IL-4 and IL-13 signaling, and immunoregulatory interactions between lymphoid and nonlymphoid cells (Fig. 4a–c and Supplementary Fig. 4–6). The enriched biological processes of the downregulated top100 genes among monocyte subclusters between cases and controls were shown in Supplementary Fig. 7, and the pathways analysis of DEGs in other immune cells was also conducted (Supplementary Fig. 8). Moreover, we noticed that cell communications between monocyte C1 subset and other cell types in PBMCs of HAPH patients tended to be relatively weak compared with that in the controls (Supplementary Fig. 9). A similar finding was observed in the monocyte C2 subpopulation (Supplementary Fig. 10).

Functionally, monocytes are known to be responsible for phagocytosis, coagulation, wound healing, antiapoptotic responses and reactions to stimuli[18]. Therefore, we analyzed how these biological processes were modulated in the different samples. Interestingly, we found that HAPH patients had lower signature scores relevant to the pathways (eg. phagocytosis, myeloid cell differentiation, coagulation, platelet adhesion to exposed collagen, and regulation of TNF production etc.) compared to that of controls independent of monocyte subsets, while more obviously in C1 (non-classical) and C2 (intermediate) monocytes (Fig. 4d). These observations suggested that HAPH patients may suffer from a severe impairment in monocyte functions. In addition, we found that the genes associated with the response to virus and IL-2 biosynthesis were markedly upregulated in C1 and C2 clusters obtained from the HAPH samples (Fig. 4d), which suggested that specific immunological responses might be triggered in these patients.

Next, we ranked the contributions of genes involved in each biological pathway in the Fig. 4d and selected 3 genes with the largest fold change in each biological pathway. A total of 15 candidate genes were determined from the 7 pathways. Firstly, we established a hypoxia-induced PH model in mice, with higher RVSP identified by right heart catheterization (Fig. 5a), right ventricular hypertrophy index (Fig. 5b), and histopathological evaluation (Fig. 5c, d). We analyzed the mRNA expression levels of 15 genes in the lung tissue of hypoxic PH mice and found that the expression of 5 genes (*Prf1*, *Spon2*, *Tbx21*, *Cd28* and *Cd3e*) were significantly increased compared to that in control mice (Fig. 5e), which were consistent with the results of scRNA-seq analysis in HAPH patients. The rest 10 genes showed no significant difference in lung tissue of hypoxic PH mice (Supplementary Fig. 11). Next, we quantitatively analyzed 5 genes at the protein level, and found that *Cd28*, *Tbx21*, and *Spon2* were significantly elevated in the lung tissues of hypoxic PH mice (Fig. 5f–i, Supplementary Fig. 12), in line with their transcriptional levels. The expression levels of *Prf1* and *Cd3e* were not effectively detected in the lung tissues of hypoxic PH mice (Supplementary Fig. 12). A cellular model of chemically induced hypoxia showed that Cd28, Tbx21, and Spon2 were significantly increased both in RAW264.7 monocytes/macrophages and THP-1

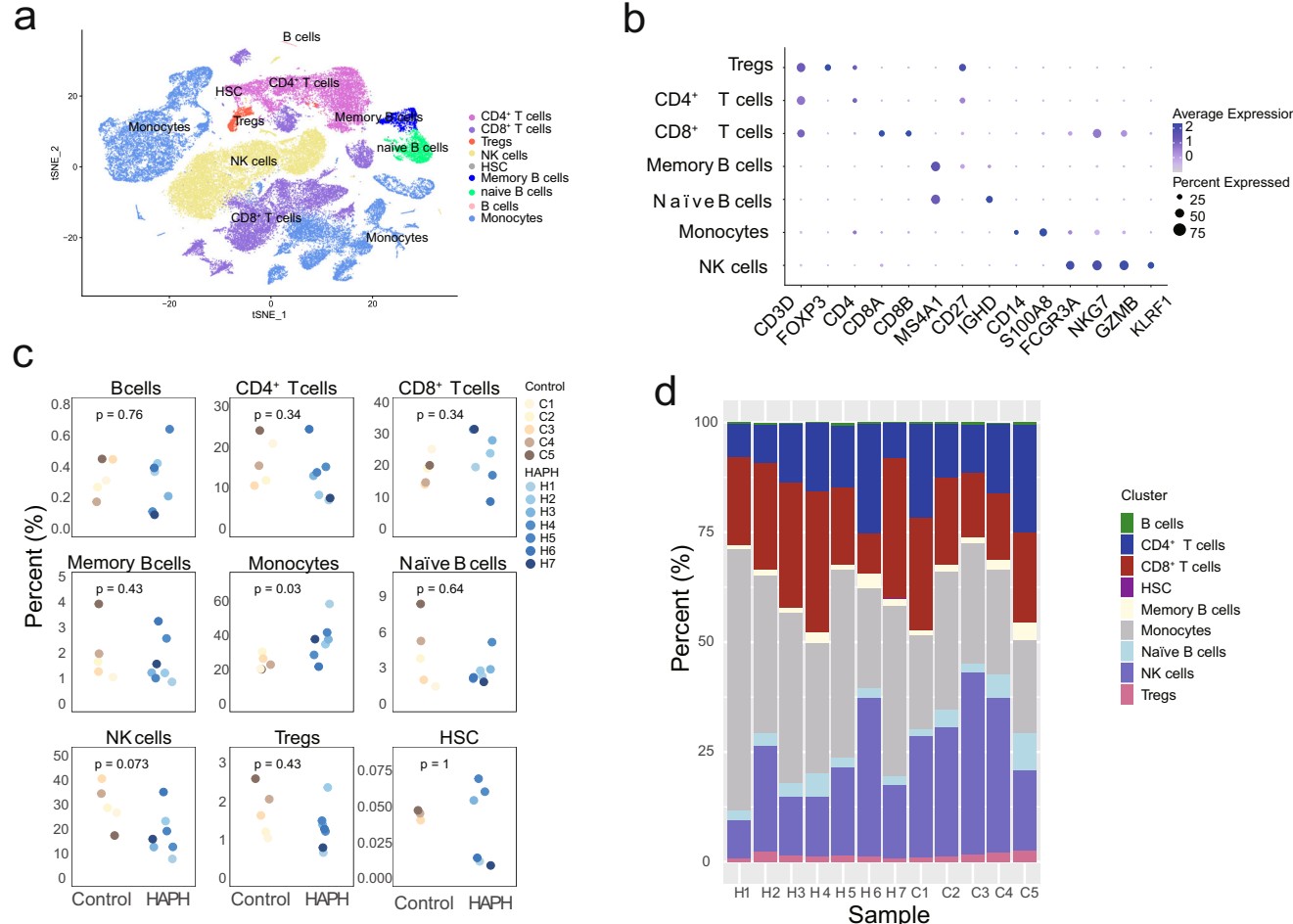

**Fig. 2 | Differential immune cell composition in the circulation of patients with HAPH. a** t-SNE plot of the main immune cell subsets. **b** Dot plot depicting the percentages and average expressions of the canonical genes associated with each main immune cell cluster. **c** Boxplots comparing the percentages of indicated cell types in PBMCs between HAPH patients (*n* = 7) and control participants (*n* = 5). The two-sided *p* values from the Wilcoxon rank-sum test were shown. **d** Proportions of each cell type in HAPH patients (*n* = 7) and control participants (*n* = 5). HAPH high-altitude pulmonary hypertension, t-SNE t-distributed stochastic neighbor embedding, PBMCs peripheral blood mononuclear cells.

monocytes induced in response to CoCl$_2$ (Fig. 5j–l, Supplementary Fig. 13). Co-cultured with Jurkat Clone E6-1 cells induced an increase in the cell viability of human PASMCs, and the knockdown of *CD28* in Jurkat Clone E6-1 cells significantly inhibited this effect (Fig. 5j, k). Neither *TBX21* nor *SPON2* knockdown exhibited similar significant inhibitory effects (Supplementary Figs. 14, 15).

### Decreased hypoxia-inducible transcription factor-1α (HIF-1α) in all monocyte subsets associated with HAPH

It is evident that HIF-1α and HIF-2α are involved in the pathophysiology of pulmonary vascular remodeling[11–13]. Therefore, we comprehensively examined whether HIFs are associated with HAPH in our investigation. According to the analysis, HIF-1α did not differ between HAPH patients and control individuals in all PBMCs, NK cells, B cells or T cells (Fig. 6a, d), while HIF-2α was not efficiently detected from all the samples. Notably, a marked downregulation of HIF-1α expression was found in the monocytes between HAPH patients and controls (Fig. 6e). Specifically, the C0 (classical) and C1 (nonclassical) monocytes in HAPH showed lower expressions of HIF-1α relative to that of controls (Fig. 6f). Meanwhile, the expression of VEGFA (an important downstream protein of HIF-1α) in HAPH patients was much lower independent of monocyte subsets in comparison of control participants, a similar trend as that of HIF-1α between the two groups (Fig. 6g–i). The correlation of clinical parameters and expression of HIF1-α and its

target gene VEGFA were shown in Supplementary Table 4. We also investigated Hif-1α expression in lung tissues of PH mice and its expression in response to hypoxia-mimetic agent. As shown in Supplementary Fig. 16 and 17, Hif-1α in lung issues of hypoxia induced mice was significantly higher than that of controls (Supplementary Fig. 16a,b and Supplementary Fig. 17a, b). At the cellular level, Hif-1α (HIF-1α) was also significantly increased both in RAW264.7 monocytes/macrophages (Supplementary Fig. 16c, d and 17c, d) and THP-1 monocytes induced in response to CoCl$_2$ (Supplementary Fig. 16e, f and 17e, f). Taken together, our data indicate that HAPH patients may suffer from impairment of HIF-1α mediated signaling in the monocytes of the circulating system.

### Comparison of immune cell compositions in PBMCs between HAPH and PH patients

To compare the immune adaptations with HAPH and PH, 6 PH patients were recruited (Supplementary Table 5). We then performed scRNA-seq on PBMCs from PH patients. The PH immune landscape was depicted following the same analysis workflow. We identified 11 cell types according to the canonical markers, including monocytes, NK cells, CD4+, CD8+, and regulatory T cells, memory and naive B cells, cDC, pDC and HSCs (Fig. 7a, b). Among them, monocytes accounted for 26.7%. The landscape comparison of HAPH and PH showed that their immune adaptions were similar, both in terms of cell type

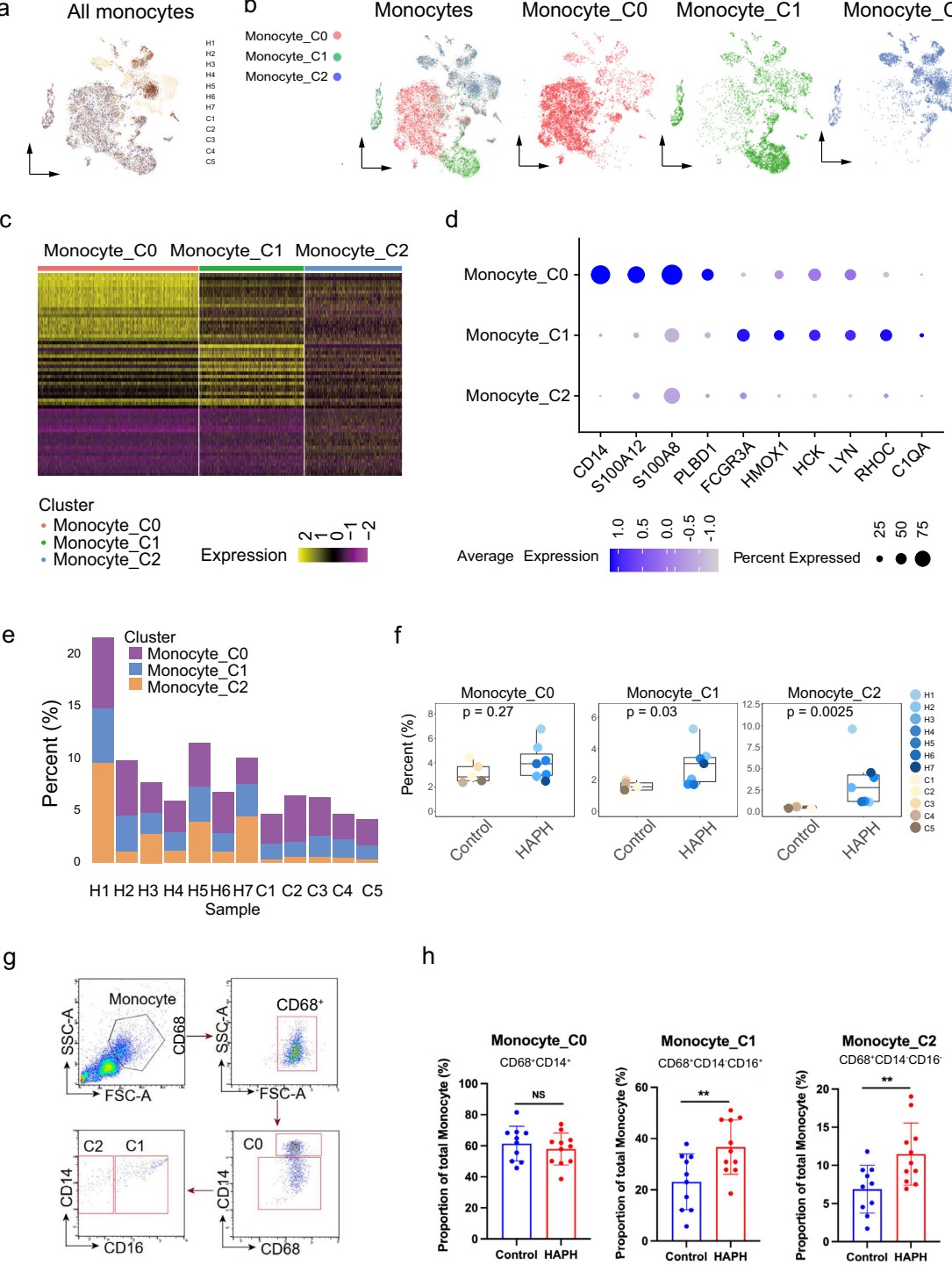

**Fig. 3 | Expansion of monocyte clusters in circulation system of patients with HAPH. a** t-SNE plot of all monocytes collected in the present study. **b** t-SNE plot of all monocytes and the sub-clusters with cells colored based upon canonical gene markers of monocytes. clusters 0 (C0) represented as classical monocytes, C1 as non-classical monocytes and C2 as intermediate monocytes. **c** Heatmap displaying the scaled expression values of discriminative gene sets from all monocytes in PBMCs between HAPH and controls. The top 50 marker genes in each subgroup were shown. **d** Violin plots indicating the expression of genes in each monocyte cluster in PBMCs from both HAPH and controls. **e** Proportions of each monocyte cluster in each sample as indicated. **f** Boxplots comparing the percentages of each monocyte cluster in PBMCs between HAPH patients ($n = 7$) and control participants ($n = 5$). The two-sided $p$ values from the Wilcoxon rank-sum test were shown. **g** Gating strategy of Monocytes by flow cytometry. **h** The comparison of the proportion of each monocyte subsets in PBMCs between HAPH patients ($n = 11$) and controls ($n = 10$) assessed by flow cytometry. (*$p < 0.05$; Unpaired $t$ test was utilized as appropriate. Data were presented as mean ± SD). HAPH high-altitude pulmonary hypertension, t-SNE t-distributed stochastic neighbor embedding, PBMCs peripheral blood mononuclear cells.

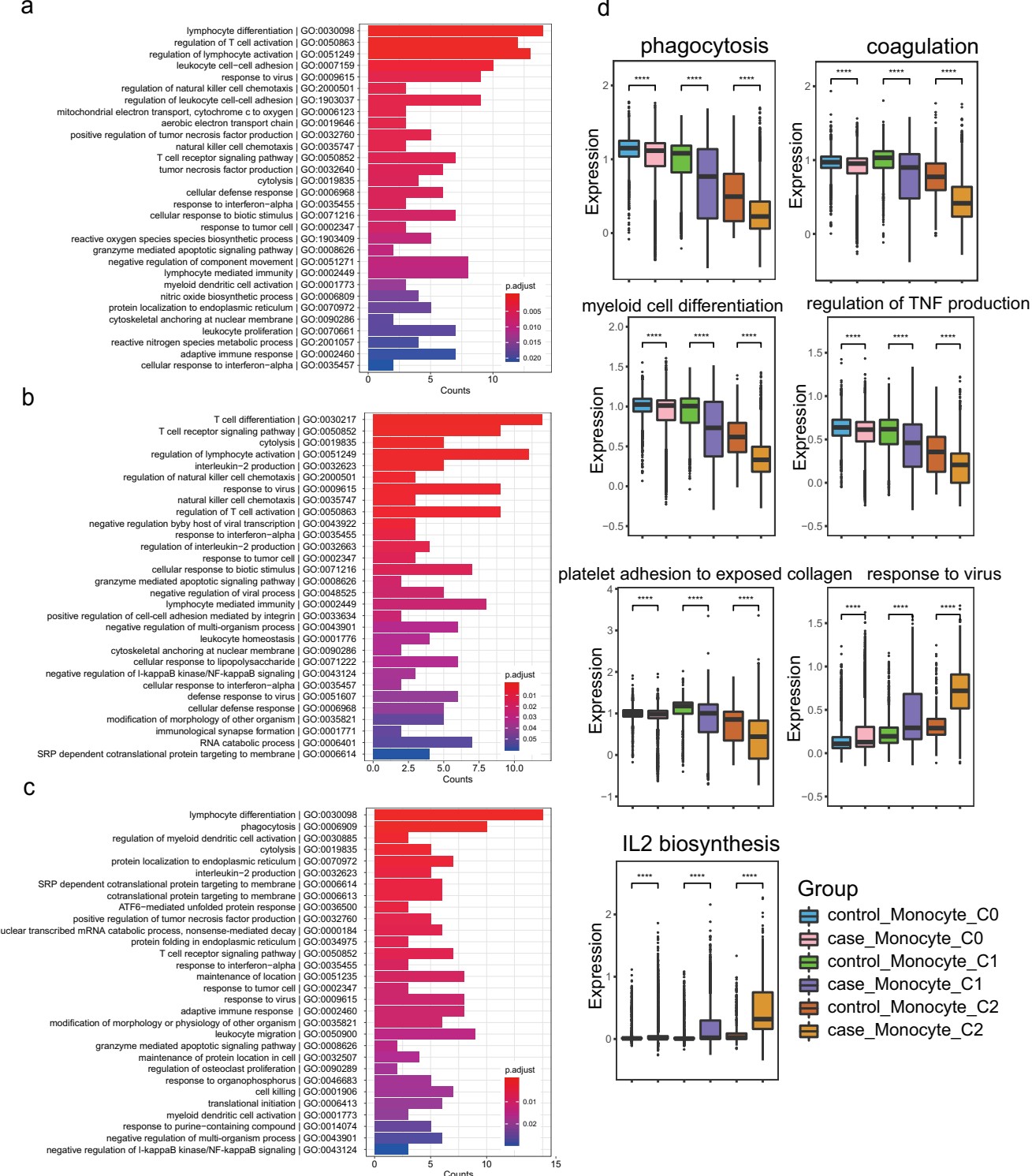

**Fig. 4 | Differential biological pathways in each monocyte cluster between HAPH patients and controls.** Gene Ontology analysis (biological process) of the DEGs in C0 (**a**), C1 (**b**) and C2 (**c**) monocytes between the HAPH and control samples. **d** Boxplot showing the mean pathway signature score of each monocyte subset from each group. ***$p < 0.001$; ****$p < 0.0001$; using unpaired Wilcox rank sum test. HAPH high-altitude pulmonary hypertension, DEGs differentially expressed genes.

composition and cell fractions in each population (Fig. 7c, d). Similar cell proportions of CD4[+] T cells, CD8[+] T cells and NK cells were documented in PH as in HAPH (Fig. 7e, f). Notably, regarding the comparison of the monocyte subsets between HAPH and PH, C2 exhibited a significantly higher proportion in HAPH, which was not the case in C0 and C1 (Fig. 7g). A single tSNE of all the samples was generated to identify clusters with the same analysis parameters (Supplementary Fig. 18), tSNE plots of individual samples were shown in Supplementary Fig. 19. Frequencies of the three monocyte subsets were compared among HAPH patients, PH patients and controls (Supplementary Fig. 20). Overall, scRNA-seq of PBMC hinted that similar immune adaptations may exist in HAPH and PH.

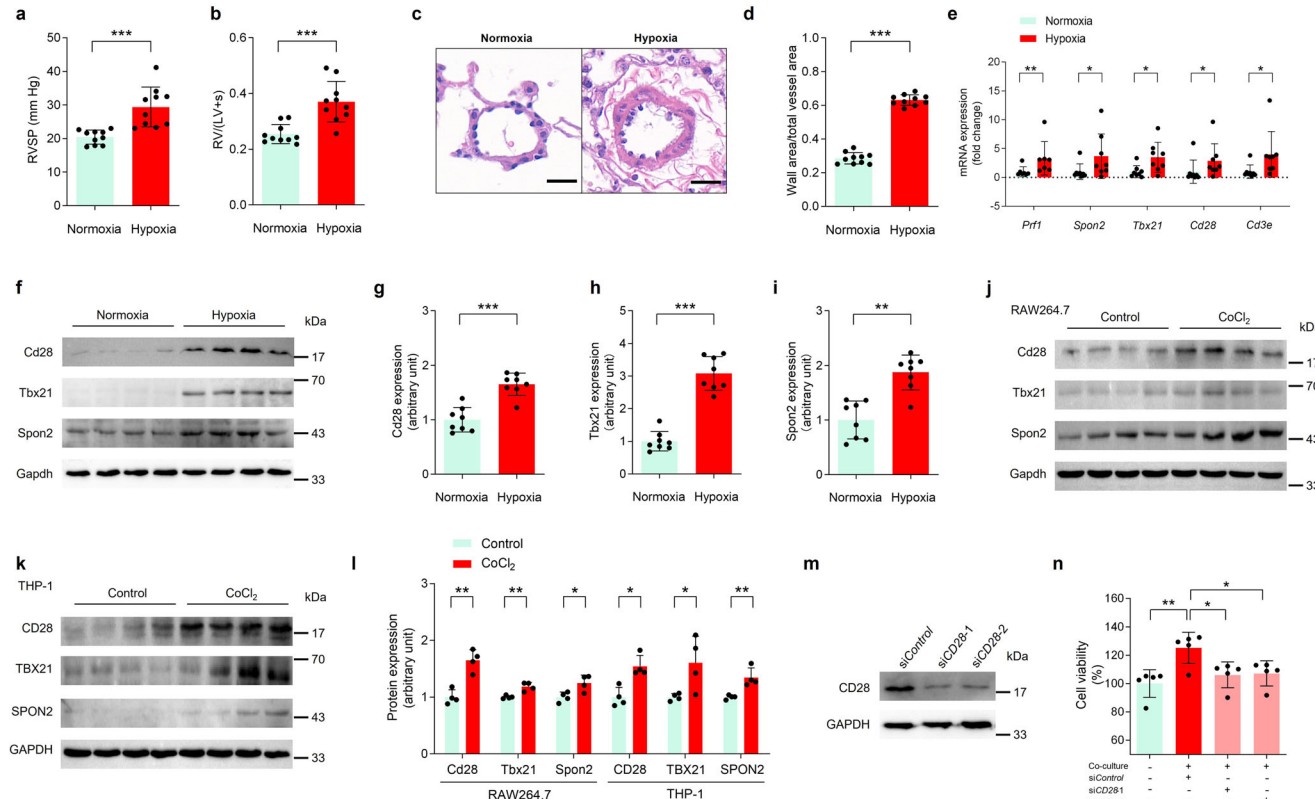

**Fig. 5 | Gene-function validation in hypoxic PH mice.** C57BL/6 mice exposed to hypoxia (10% $O_2$) for 4 weeks exhibited higher RVSP (**a**) and RVHI (**b**) compared to that of mice in normoxia (21% $O_2$) ($n = 10$ for per group, ***$p < 0.001$; Student's $t$ test. Data were presented as mean ± SD). **c** Representative images of hematoxylin and eosin (H&E) staining of lung tissues from mice under normoxic or hypoxic condition. Scale bars, 20 μm. **d** Assessment of pulmonary vascular remodeling by determination of the ratio of the media cross-sectional area to the total vessel cross-sectional area in each group ($n = 10$ for per group, ***$p < 0.001$; Student's $t$ test. Data were presented as mean ± SD). **e** The expression of *Prf1*, *Spon2*, *Tbx21*, *Cd28* and *Cd3e* in lung tissues of the mice under normoxic or hypoxic condition at mRNA levels ($n = 7$–8 for per group, *$p < 0.05$, **$p < 0.01$; Student's $t$ test. Data were presented as mean ± SD). **f**–**i** Representative images of immunoblottings and the quantification of the expression of Cd28 (**g**), Spon2 (**h**) and Tbx21 (**i**) in lung tissues of the control mice and hypoxic PH mice at protein levels ($n = 8$ for per group, **$p < 0.01$, ***$p < 0.001$; Student's $t$ test. Data were presented as mean ± SD).

**j** Immunoblottings for the expression of Cd28, Tbx21 and Spon2 in RAW264.7 monocytes/macrophages in response to CoCl2 (150 μmol/L) or vehicle for 24 h. **k** Immunoblottings for the expression of CD28, TBX21 and SPON2 in THP-1 monocytes in response to CoCl2 (150 μmol/L) or vehicle for 24 h. **l** Quantification of the expression of Cd28 (CD28), Tbx21 (TBX21) and Spon2 (SPON2) in RAW264.7 and THP-1 cells ($n = 4$ for per group, *$p < 0.05$, **$p < 0.01$; Student's $t$ test. Data were presented as mean ± SD). **m** Representative images of immunoblottings of Jurkat Clone E6-1 cells transfected with CD28 siRNA (si*CD28*) or control siRNA (si*Control*). **n** Viability of human PASMCs co-cultured with or without *CD28*-silencing or control Jurkat cells for 48 h ($n = 5$ for per group, *$p < 0.05$, **$p < 0.01$; one-way ANOVA, Tukey's post-hoc test. Data were presented as mean ± SD). PH pulmonary hypertension, RVSP right ventricular systolic pressure, RVHI right ventricular hypertrophy index, PASMCs pulmonary arterial smooth muscle cells. Prf1 Perforin 1, Spon2, Spondin 2, Tbx21 t-box transcription factor 21.

## Discussion

An increasing body of evidence suggests that disordered immune and inflammatory responses would culminate in the pathogenesis of PH. However, there are limited studies that have investigated the circulating immune landscape in patients with HAPH (third category of PH). By profiling the gene signature of single immune cell in PBMCs from HAPH patients or control individuals, we identified and characterized a distinct monocyte phenotype in the peripheral circulation in association with HAPH.

A main finding from our investigation is that monocytes were more prevalent in PBMCs from patients with HAPH than in those from the controls. Indeed, previous studies have tried to clarify how inflammation is present in the human pathological process of PH. However, no consensus has been reached regarding how monocytes are involved, especially in HAPH. Marsh et al observed an increase in lymphoid cells and decrease in myeloid lineages (e.g., neutrophils and monocytes) in PH lung tissues, while in isolated pulmonary arteries, increased abundances of activated macrophages and monocyte populations were detected[24]. Experimentally, blocking the recruitment of circulating monocytes may help to protect against schistosomiasis-

induced PH[25]. Here, we presented evidence that monocyte abundance was higher in the circulation system of HAPH patients.

As important mediators of the innate immune system, monocytes consist of several subpopulations that are generated through distinct developmental pathways and differ in their inflammatory and migratory phenotypes[18,26,27]. In the present study, we reported an intriguing observation that HAPH patients exhibited a distinct monocyte phenotypic alteration compared to that of controls, as evident by a higher abundance of C1 (non-classical) and C2 (intermediate) monocytes rather than C0 (classical) monocytes in PBMCs from HAPH patients. Moreover, all three monocyte subsets exhibited significantly decreased levels of phagocytosis, myeloid cell differentiation, coagulation, and platelet adhesion to exposed collagen, which suggests a functional impairment in all monocyte subsets in HAPH patients.

It is widely known that classical monocytes are bone-marrow derived, highly phagocytic and can be rapidly recruited to sites of inflammation via the activity of the chemokine receptor, CCR2[28]. Previously, Yen-Rei et al. found that although classical monocyte recruitment and accumulation can be abrogated by CCR2 deletion, PH was not alleviated and became even more serious in CCR2-deficient mice,

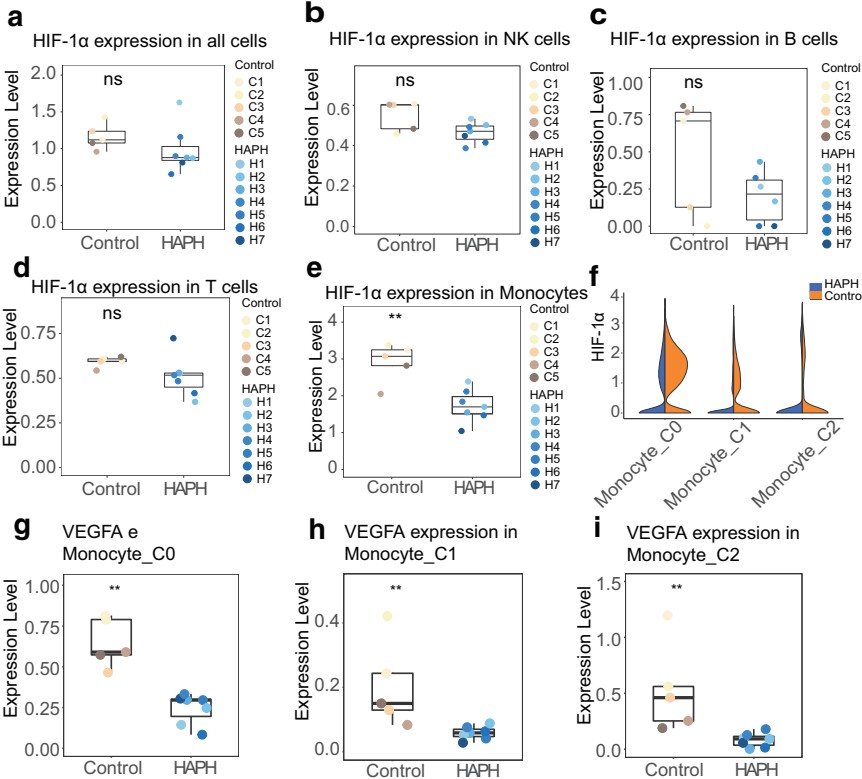

**Fig. 6 | Comparison of the expression of HIF-1α and VEGF1 in immune cell population between HAPH and controls.** Box plots displaying the expression of HIF-1α in HAPH patients relative to that in controls in indicated cell population (**a**: all cells, **b**: NK cells, **c**: B cells, **d**: T cells, and **e**: monocytes). **\*\*** $p < 0.01$; n.s. not significant by unpaired two-sided $t$ test. **f** Box plots showing the comparison of the expression of HIF-1α in each monocyte cluster between HAPH patients ($n = 7$) and controls ($n = 5$). **g–i** Box plots showing the comparison of the expression of VEGFA in each monocyte cluster between HAPH patients ($n = 7$) and controls ($n = 5$). HIF-1α hypoxia-inducible factor-1α, VEGF vascular endothelial growth factor, HAPH high-altitude pulmonary hypertension, NK cells natural killer cells.

which strongly suggested that classical monocytes may not directly contribute to the development of PH[29]. This was in accordance with our data, which showed that C0 (classical) monocytes were not significantly elevated in HAPH patients.

Most recently, nonclassical monocytes were displayed to recruit to small pulmonary arteries, differentiate into pulmonary interstitial macrophages, and promote vascular remodeling afterward[14]. We showed that circulating non-classical monocytes were consistently more frequent in HAPH patients than in the controls. Non-classical monocytes, which comprise only approximately 2–11% of all circulating monocytes, are also bone-marrow derived. They are mobile and can patrol the microvasculature by crawling on the endothelium and infiltrate into subendothelial spaces in response to inflammatory stimuli[30]. Therefore, the above findings suggest that non-classical monocytes might participate in HAPH development.

At present, the biological relevance of intermediate monocytes during PH remains to be determined. In our single-cell transcriptome analyses, we observed a marked increase in circulating intermediate monocyte subset in HAPH patients, which has not been reported previously. Intermediate monocytes represent a minor subset of blood monocytes, and their functions involve reactive oxygen species (ROS) production, antigen presentation, cytokine secretion and apoptosis regulation[31]. Interestingly, intermediate monocytes appear to be relevant for the healing phase after myocardial injury in the heart and act as an independent predictor for cardiovascular events[32–34]. It was reported that humans with hypertension have increased levels of intermediate and non-classical monocytes and that NO may inhibit the formation of intermediate monocytes[35]. In our analyses, the intermediate monocytes were remarkably increased in HAPH patients, which were accompanied by downregulated pathway signatures such

as phagocytosis, coagulation, and platelet adhesion to exposed collagen. However, how this subpopulation participates in the pathogenesis of PH still needs to be further explored in future studies.

Furthermore, another intriguing finding in our study is that HIF-1α expression was drastically downregulated in PBMCs from patients with HAPH compared with that of the control, especially in all monocyte subsets, which suggests a tight connection between hypoxia and inflammatory responses. Previous reports have addressed the participation of HIFs in hypoxia-induced PH development (e.g., WHO group 3 PH) in animal models[36, 37]. HIFs are α/β heterodimeric transcription factors that can transactivate various genes that may help protect against the consequences of hypoxia. Although the best-characterized HIFα isoforms, HIF-1α and HIF-2α, bind to an identical core consensus, they have disparate functions in different cell types[38, 39]. HIF-2α, instead of HIF-1α, is found to drive the development of PH, since inhibition of HIF2α signaling attenuates the initiation of hypoxia-induced PH[36]. HIF-1α is widely expressed and detected in virtually all innate and adaptive immune populations. Myeloid HIF-1α is known to contribute to inflammation, as bacterial infection induces HIF-1α expression in human monocytes[40]. The inhibition of HIF-1α-blocked monocyte induction of trained immunity refers to the capacity of the innate immune system to build immunological memory by metabolic or epigenetic reprogramming[41]. Therefore, our data that showed downregulation of HIF-1α expression in peripheral monocytes in HAPH patients might suggest an impaired function of monocytes in response to hypoxia. It was reported that immunity-trained monocytes might respond to intruding pathogens in a robust and quicker manner in terms of proinflammatory cytokine production and possibly through enhanced phagocytosis capacity[42]. This observation was in

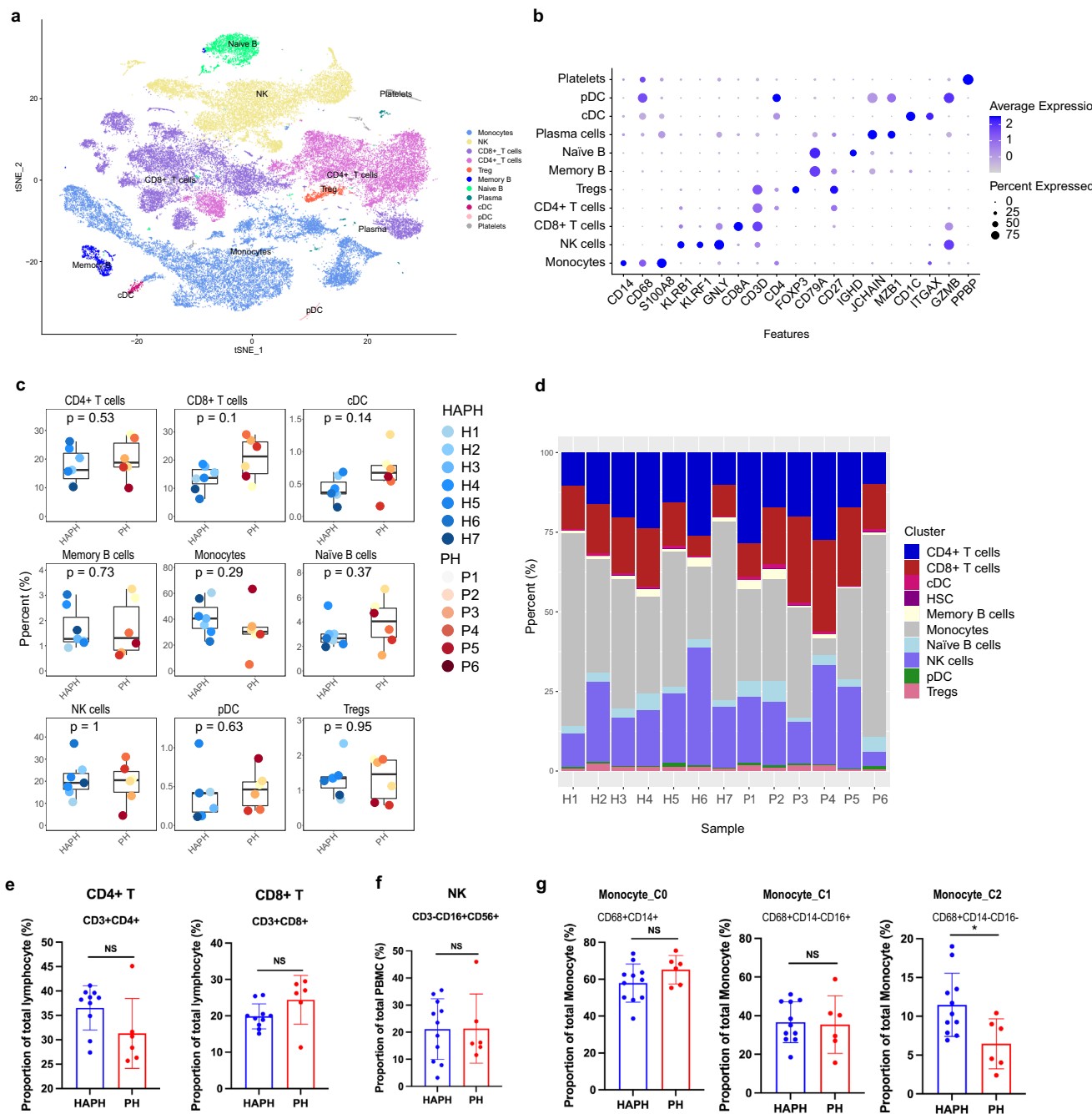

**Fig. 7 | Immune cell composition in the circulation system of patients with PH and its comparison with those in HAPH. a** t-SNE plot of the main immune cell subsets in PBMCs from patients with PH ($n = 6$). **b** Dot plot depicting the percentages and average expressions of the canonical genes associated with each main immune cell cluster in PBMCs from patients with PH ($n = 6$). **c** Boxplots comparing the percentages of each main cell type in PBMCs between HAPH ($n = 7$) and PH ($n = 6$) patients. The two-sided $p$ values from the Wilcoxon rank-sum test were shown. **d** Proportions of each cell type in each sample as indicated. (**e–g**) The proportion of T cells, NK cells and C0, C1, C2 monocyte subsets in PBMCs from HAPH ($n = 11$) and PH ($n = 6$) patients as analyzed by flow cytometry (*$p < 0.05$; Unpaired $t$ test was utilized as appropriate. Data were presented as mean ± SD). PH pulmonary hypertension, HAPH high-altitude pulmonary hypertension, t-SNE t-distributed stochastic neighbor embedding, PBMCs peripheral blood mononuclear cells.

accordance with our data. One of the possibilities of the discrepancy might lie in the fact that the augmentation of HIF-1α in cells with an enhanced phagocytosis was induced by pharmacological stress (short period and acute) in the former one, while long-term (chronic) hypoxia exposure at high altitude might cause monocyte exhaustion, manifested with impaired HIF-1α mediated signaling pathways (eg. phagocytosis), It could also be postulated that the impairment of HIF-1α mediated function might in turn facilitate the monocyte expansion or mobilization from bone marrow into circulation to adapt long-term hypoxia challenge, reflective of a higher frequency of circulating non-classical and intermediate monocytes in HAPH patients.

There are several limitations in our investigation. The sample size was relatively small, and only peripheral blood was assessed. Although our study demonstrated some intriguing findings, future in vivo and in vivo studies will be needed to clarify the mechanism underlying the impaired HIF-1α mediated function in monocytes and their roles in PH.

In summary, we applied single-cell transcriptomics to characterize the circulating immune responses in HAPH patients. We determined that intermediate and non-classical monocytes were more frequent in HAPH patients, concomitant with dramatic changes in the monocyte phenotypes of all monocyte subsets, including HIF-1α downregulation and impaired gene expression in the phagocytosis, coagulation and wound healing pathways. This work highlighted a link between circulating monocyte changes in response to hypoxia and the pathophysiology of HAPH development.

## Methods

### Participants and clinical characteristics

This study was approved by the Ethics Committees of the First Affiliated Hospital of Dali University and Peking Union Medical College Hospital, and the relevant informed consent documents were signed by the participants before sample collection and data acquisition, all participants received no compensation from this study. In First Affiliated Hospital of Dali University, a total of 248 Naxi villagers were selected from Gaomei Village (3194.97 m), Mushu Village (3109.72 m) and Lutu Village (2981.71 m) in Qihe Town, Lijiang City, Yunnan Province. 11 patients with systolic PAP (sPAP) ≥40 mmHg were finally considered as HAPH patients and recruited in our study. At the same time, 15 contemporary individuals who lived at the same altitude and were not relatives within three generations were recruited as controls. In Peking Union Medical College Hospital, 6 PH cases living at 45 m altitude were also enrolled. Candidates with left-heart disease, lung disease, and renal insufficiency were excluded. We conducted scRNA-seq analysis comparing between 7 HAPH and 5 controls, and also between 7 HAPH and 6 PH. Validation studies in PBMCs were carried out among 6 PH,11 HAPH and 10 controls with fluorescence-activated cell sorting (FACS) analysis (Fig. 1).

In this study, standardized questionnaires regarding cardiovascular risk factors, including basic information about age, sex, smoking status, cholesterol, diabetes, symptoms, previous medical history and personal history, were recorded. Fasting blood samples were drawn for the analysis of blood tests including cell counts, glucose, cholesterol, blood urea nitrogen and creatinine. All echocardiography parameters were collected by the same experienced cardiologist. The sPAP was calculated according to the maximum tricuspid regurgitation velocity.

### Isolation of PBMCs and preparation of single-cell suspensions

The PBMCs were isolated from anticoagulated blood according to the Histopaque-1077 procedure (Sigma-Aldrich Co. LLC). Briefly, 3 mL of whole blood were carefully layered onto 3 mL of Histopaque-1077 in a 15-ml conical centrifuge tube and then centrifuged for 30 min at $700 \times g$ at room temperature without braking. After centrifugation, the upper layer above the opaque interface containing mononuclear cells was removed with a Pasteur pipette. The opaque interface was carefully transferred into a clean conical centrifuge tube, washed by adding 10 mL of 1x phosphate-buffered saline (PBS) supplemented with 0.04% bovine serum albumin (BSA) and mixed, followed by centrifugation at $300 \times g$ for 5 min at 4 °C. The supernatant was aspirated and discarded, and the cell pellet was washed twice. After that, the cell pellet was resuspended in 0.5 mL of 1× PBS containing 0.04% BSA. For each sample, the cell numbers were counted, and the cell viability was evaluated by CountStar.

### Droplet-based single-cell sequencing

Using the Single Cell 5' Library and Gel Bead Kit (10X Genomics, PN-1000165) and Chromium Single Cell G Chip Kit (10X Genomics, PN-1000120), the cell suspension was loaded onto a chromium single-cell controller (10X Genomics) to generate single-cell gel beads in the emulsion (GEMs) according to the manufacturer's protocol. Briefly, cell pellets were suspended in PBS containing 0.04% BSA.

Approximately 10,000 cells were added to each channel, and approximately 6000 cells were recovered. The captured cells were lysed, and the released RNA was barcoded via reverse transcription in individual GEMs. Reverse transcription was performed at 53 °C for 45 min and was followed by 85 °C for 5 min, after which the temperature was held at 4 °C. Complementary DNA (cDNA) was generated and amplified, and its quality was assessed using an Agilent 4200 system (performed by CapitalBio Technology, Beijing) according to the manufacturer's instructions. The barcoded sequencing libraries were generated using Chromium Single Cell 5' Reagent Kits (10X Genomics, PN-1000080) and were sequenced across 8 lanes on a NovaSeq 6000 platform targeting 100,000 reads per cell with a paired-end 150 bp pair-end (PE150) mode (performed by CapitalBio Technology, Beijing).

### Single cell RNA-seq (scRNA-seq) data processing

The sequencing data were aligned to the human reference genome (GRCh38) and processed using CellRanger (version 4.0.0, https://support.10xgenomics.com/single-cell-gene-expression/software/pipelines/latest/what-is-cell-ranger). The gene expression matrix obtained from the CellRanger pipeline was filtered and normalized using the Seurat R package (version 3.2, https://cran.r-project.org/web/packages/Seurat/index.html)[43]. Cells were selected if they met the following criteria: (i) top 99% of cells in unique molecular identifier (UMI) counts; (ii) >200 genes; and (iii) <25% of mitochondrial gene expression in the UMI counts. After removal of low-quality cells, the gene expression matrices were normalized to the total UMI counts per cell and were transformed to a natural log scale. Then, all datasets from the individual samples were integrated using the "FindIntegrationAnchors" and "IntegrateData" functions in Seurat. We identified "anchors" among all individual datasets with multiple canonical correlation analysis (CCA) that was implemented by the "FindIntegrationAnchors" function and used these "anchors" in the "IntegrateData" function to create a batch-corrected expression matrix of all cells, which allowed the cells from different datasets to be integrated and analyzed.

The integrated data were scaled to a unit variance and zero mean, and the dimensionality was reduced by principal component analysis (PCA). A shared nearest neighbor (SNN) graph was constructed based on the distances in PCA space (top 1:50 principal components) using the "FindNeighbors" function, in which the k-nearest neighbors (KNN) of each cell were first determined and this KNN graph was then used to construct the SNN graph by calculating the neighborhood overlap (Jaccard index) between each cell and its KNN. The Louvain algorithm was applied to iteratively group proximal cells together with the "FindClusters" function with a resolution of 0.6. Visualization was achieved by the t-distributed stochastic neighbor embedding (tSNE) projection. Cell type was annotated via SingleR[44], along with marker-based annotations.

### Differential expression and functional enrichment analysis

After dimensional reduction and projection of all cells into two-dimensional space by tSNE, the cells were clustered together according to their common features. The "FindAllMarkers" function in Seurat was used to find the markers for each of the identified clusters. For each cluster, we performed functional enrichment analysis, which was implemented by clusterProfiler (version3.10.1, https://bioconductor.org/packages/release/bioc/html/clusterProfiler.html) using the top 100 DEGs sorted by logFC in ascending order and p.adj ≤0.05. The enrichment analysis of comprehensive functions included Gene Ontology (GO), Kyoto Encyclopedia of Genes and Genomes (KEGG) pathways, Reactome and Disease. Gene set enrichment analysis was performed by the GSEA application version of JAVA (version 2.2.2.4, https://www.gsea-msigdb.org/gsea/index.jsp) with predefined gene sets from the Molecular Signatures Database (MSigDB, version 6.2, https://www.gsea-msigdb.org/gsea/msigdb/human/collections.jsp).

## Determination of the major cell types

Cell type annotations were performed according to the Blueprint and Encode reference dataset via SingleR, which assigns cellular identity for single cell transcriptomes by comparison to reference datasets of pure cell types sequenced by microarray or RNA-seq. In detail, we corrected the annotations of SingleR according to the following: (i) "Class-switched_mem", "Class-switched memory B-cells" and "Memory_B-cells" were defined as "Memory_B-cells"; (ii) "CD8⁺_T-cells", "CD8⁺_Tcm" and "CD8⁺_Tem" were defined as "CD8⁺_T-cells"; and (iii) "CD4⁺_T-cells", "CD4⁺_Tcm" and "CD4⁺_Tem" were defined as "CD4⁺_T-cells". In total, we classified all cells into five major cell types, including T cells (e.g., CD4⁺ T cells, CD8⁺ T cells and Tregs), B cells (memory B cells, and naïve B cells), NK cells, monocytes, and hematopoietic stem cells (HSCs).

## Cell−cell communication analysis

To explore cell-cell communications via ligand−receptor interactions, we employed the strategy that was proposed by Vento-Tormo et al. based on CellPhoneDB (version 2.0, https://www.cellphonedb.org/) database[45], a public repository of ligands, receptors and interactions. The interaction scores between two cell types are mediated by a specific ligand-receptor pair based on the average gene expression of the ligand from one cell type and the corresponding receptor from another cell type. To identify the significant cell-cell interactions, we permuted the change of cell type label for each cell 1000 times to calculate the significance of each pair ($p$-value <0.01). This procedure was performed between all pairs of cell types. The network plot was drawn using iTALK (version 0.1.0, https://github.com/Coolgenome/iTALK).

## Animal models

All procedures involving mice were approved by the Animal Care and Use Committee of Peking Union Medical College Hospital, Chinese Academy of Medical Sciences. The experiment was carried out in accordance with the Guideline for Care and Use of Laboratory Animals published by the US National Institutes of Health, and the Guidelines for the ethical review of laboratory animal welfare People's Republic of China National Standard GB/T 35892-2018[46].

In total, 20 adult C57BL/6 mice (10 males and 10 females, 8 weeks old, 18−22 g) were purchased from Charles River Laboratories (Beijing, China) and allowed to acclimate for 5 days in a specific pathogen-free grade barrier system with alternating 12 h light/dark cycles at a relative humidity of 50% ± 5% with 25 °C. The mice were placed in a hypoxic chamber containing 10% $O_2$ after acclimatization. The chamber was opened three times a week for cleaning and replenishment of food and water. Soda lime was used to reduce the concentration of carbon dioxide. For normoxic condition, mice were kept in the same room with the same 12-h-light-12-h-dark cycle. The animals were sacrificed 28 days after hypoxia exposure or in the room air.

## Hemodynamic measurement, tissue collection and morphometric analysis

For measurement of right ventricular systolic pressure in mice, a fluid-filled catheter (Millar, SPR-671NR) was inserted into the right ventricular through the right jugular vein and connected to a force transducer as reported previously[47]. Mice were then sacrificed, and the hearts and lungs were harvested. Right ventricular hypertrophy was determined as a ratio of the weight of right ventricle to the total weight of left ventricle and septum (RV/LV + S). Lung tissues were harvested and stored immediately at −80 °C or in formalin for subsequent analysis.

A fraction of the excised lungs was fixed in 4% paraformaldehyde, embedded in paraffin and then sliced into 4-μm-thick sections. Sections were stained with hematoxylin and eosin (H&E) following the standard protocol for histological examination as in previous study. Vessel remodeling was calculated as (the external vessel area−the internal vessel area)/the external vessel area, as described previously[8].

## Real-time PCR

Total RNA was isolated from frozen mouse lung tissues with Trizol (#15596026, Invitrogen), and then reverse transcribed into cDNA using a SuperScript III First-Strand Synthesis Kit (#18080-051; Thermo Fisher Scientific) according to the manufacturer's instructions. FastStart Universal SYBR Green Master (#04913850001; Roche, Basel, Switzerland) was used to quantify the PCR amplification products by a fluorescence ration PCR instrument (CFX96, Bio-Rad). Relative mRNA expression levels of target genes were normalized to *Actb* expression and calculated using the 2(−ΔΔCT) method. Primer pairs were listed in Supplementary Table 6.

## Western blotting

A total of 20 μg protein extract from mouse lung tissues or samples was resolved by 10% or 12% SDS-polyacrylamide gel electrophoresis, transferred onto a polyvinylidene fluoride membrane, and blocked with 5% BSA in Tris-buffered saline (TBS) for 1 h at room temperature. The membrane was then washed three times in TBS with 0.1% Tween 20 (TBS-T) and incubated overnight at 4 °C with anti-Spon2 antibody [EPR9799] (abcam, #ab171955, 1:1000 dilution), anti-Prf1 antibody [E7D8R] (Cell Signaling Technology, #62550, 1:1000 dilution), anti-Tbx21 antibody [E4I2K] (Cell Signaling Technology, #97135, 1:1000 dilution), anti-Cd28 antibody [D2Z4E] (Cell Signaling Technology, #38774, 1:1000 dilution) or anti-Cd3e antibody [CD3-12] (Cell Signaling Technology, #4443, 1:1000 dilution) or anti-Hif-1α [EPR16897] (abcam, #ab179483, 1:1000 dilution). The membranes were washed three times with TBS-T and then incubated with anti-rabbit or anti-mouse IgG conjugated with horseradish peroxidase in TBS. After three times washes, the membranes were reacted with a chemiluminescence system for 1 min and then exposed to X-ray film. As a loading control, all blots were reprobed with an anti-glyceraldehyde-3-phosphate dehydrogenase (Gapdh) monoclonal antibody [D16H11] (Cell Signaling Technology, #5174, 1:1000 dilution). Protein expression levels were quantified by scanning densitometry using the Quantity One 1-D analysis software (version 4.6.6, PC).

## Cell experiments

Mouse RAW264.7 monocytes/macrophages, human THP-1 monocytes and human Jurkat Clone E6-1 T cells were purchased from Procell Life Science & Technology Co., Ltd. (#CL-0190, #CL-0233, #CL-0129; Wuhan, China) and maintained according to manufacturer's instructions. The RAW264.7 and THP-1 cells were respectively seeded in 6-well plate and treated with cobalt chloride (CoCl₂; #F2116353; Shanghai Aladdin Biochemical Technology Co., LTD, Shanghai, China) at the concentration of 150 μmol/L or vehicle for 24 h. The cells were collected to detect the expression of Hif-1α, Cd28, Tbx21 and Spon2 by western blotting described above. The cells were seeded in 6-well plate and transfected with *CD28* siRNA (si*CD28*), *TBX21* siRNA (si*TBX21*), *SPON2* siRNA (si*SPON2*) or *Control* siRNA (si*Control*) using Lipofectamine 2000 (Thermo Fisher Scientific, Waltham, USA), according to the manufacturer's instructions (Supplementary Table 7). 6 h after transfection, the medium was replaced with serum-free RPMI-1640 and incubated at 37 °C with 5% CO₂ for 42 h. Cell pellets were collected for the verification of gene knockdown efficiency by western blotting. Human pulmonary artery smooth muscle cells (PASMCs) were purchased from ScienCell Research Laboratories and maintained as in our previous study[48]. The co-culture experiment of the T cells and smooth muscle cells was carried out as in previous publication[49]. The PASMCs were seeded at $1 \times 10^4$ cells per well with complete medium in 96-well

plate and incubated at 37 °C with 5% $CO_2$ for 24 h. The cells were then serum starved in DMEM/F12 for another 24 h. Culture supernatants were discarded and replaced with siCD28 or siControl transfected Jurkat Clone E6-1 T cells supplemented with 0.5% FBS and incubated at 37 °C with 5% $CO_2$ for 48 h. Cell viability was measured with MTT (3-(4,5-dimethylthiazol-2-yl)−2,5-diphenyltetrazolium bromide) assay as described previously[50].

## Flow cytometry

Peripheral blood was extracted from patients of HAPH and healthy donors and then subjected to density gradient centrifugation (Ficoll-Hypaque, TBD Science, #LTS10771, Tianjin, China) at $700 \times g$ for 20 min at room temperature. The cells at the Ficoll interface were collected, diluted in PBS, and centrifuged at $500 \times g$ rpm for 8 min. The cells were then resuspended in flow cytometry wash buffer (2% FBS in PBS) for staining according to standard protocols using the following antibodies: CD3-FITC [HIT3a] (BD Biosciences, #555339, 1:20 dilution), CD4-APC-CY7 [SK3] (BD Biosciences, #341115, 1:40 dilution) and CD8-PerCP [SK3] (Biolegend, #344708, 1:40 dilution) for T lymphocyte labeling. CD16-PE-CF594 [3G8] (BD Biosciences, #562293, 1:40 dilution) and CD56-PE-CY7 [Y1/82 A] (Biolegend, #333816, 1:40 dilution) for NK cell labeling. CD16-PE-CF594, CD68-PE-CY7 and CD14-PE [M5E2] (Biolegend, #301806, 1:40 dilution) for monocytes labeling. HIF1α – FITC [546-16] (Biolegend, #359708, 1:40 dilution), VEGFA primary antibody [EP1176Y] (Abcam, #ab52917, 1:30 dilution) and Goat Anti-Rabbit IgG H&L -AF405/DAPI (Abcam, #ab175652, 1:2000 dilution) as the secondary antibody were used for labeling intracellular expression of HIF1α and VEGFA respectively. The gating strategy employed to quantify frequencies of subcellular populations was as previously described[51]. Leukocyte subsets were analyzed using the following combination of surface markers: T cells (CD45$^+$CD3$^+$), NK cells (CD3$^-$CD16$^+$CD56$^+$), total monocytes (CD68$^+$), CD4$^+$T cells were classified as CD3$^+$CD4$^+$, CD8$^+$T cells as CD3$^+$CD8$^+$, classical monocytes (C0) as CD68$^+$CD14$^+$, non-classical monocytes (C1) as CD68$^+$CD14$^-$CD16$^+$ and intermediate monocytes (C2) as CD68$^+$CD14$^-$CD16$^-$. CD68-PE-CY7, HIF1α-FITC and VEGFA-AF504/DAPI were labeled intracellularly after permeabilization with FOXP3 Fix/Perm Buffer Set (Biolegend, San Diego, CA, USA). Data were acquired on a FACSCalibur flow cytometer (BD Biosciences, USA) and analyzed using Flow Jo software (version 10.4).

## Statistical analysis

Data were presented as the mean ± SD. Comparisons between two groups were performed using the Student's $t$ test if data were distributed normally. Otherwise, the Mann–Whitney U test was used. Comparisons of more than three groups were performed by analysis of variance (ANOVA) and Tukey's post-hoc test with GraphPad Prism version 5.0 or the Kruskal–Wallis test, as appropriate. $p < 0.05$ was considered as statistically significant.

## Reporting summary

Further information on research design is available in the Nature Portfolio Reporting Summary linked to this article.

## Data availability

The data supporting the findings from this study are available in the main manuscript and supplementary materials. The raw sequence data reported in this paper have been deposited in the GenomeSequence Archive of the Beijing Institute of Genomics (BIG) Data Center, BIG, Chinese Academy of Sciences (GSA-Human: HRA002501) and are publicly accessible at https://ngdc.cncb.ac.cn/gsa-human/browse/HRA002501[52, 53]. Any other raw data or non-commercial material used in this study are available from the corresponding author upon request. Source data are provided with this paper.

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

## Acknowledgements

We gratefully thank the patients and families for their involvement in this study. We also thank Beijing LabWecome Technology Co., Ltd. for their support and assistance in flow cytometry experiments. This work was supported by grants from the CAMS Innovation Fund for Medical Sciences (2021-I2M-1-018), National Key Research and Development Program of China (2022YFC2703902), National High Level Hospital Clinical Research Funding (2022-PUMCH-B-099), Projects of National Natural Science Foundation of China (82241020, 82170058, 81560073, 82200065, 82241007), Science Foundation for Outstanding Young Scholars of Henan Province (212300410027), Shanghai Pujiang Program (22PJ1410100), Joint Fund of Science and Technology R&D Plan of Henan Province (222103810055), Young Talent Program of Shanghai Municipal Health Commission (2022YQ070), and High-level scientific and technological talents and innovation team Program of Yunnan Province (202305AS350027) .

## Author contributions

X-H.W., Y-Y.H. and Z-R.C. designed, performed, analyzed and interpreted most experiments. Z-Y.H., G-M.W., Y.D., Y.Yang., Q-Y.Z., X-D.Y., L-Y.W., C-J.F., M.H. and J-F.F. established the clinical discovery cohort and participated in the sample collection. Y.Yan participated in data analysis, figure preparation, and revision of manuscript. Y.H. established the clinical validation cohort, collected the samples, and participated in data analysis of scRNA-seq. Y-M.S. and Y-H.R. assisted in scRNA-seq and flow cytometry experiments, and analyzed the corresponding data. S-J. Z. established the animal model and assessed the phenotypes. H.L. proposed the concept, designed, analyzed and interpreted experiments and wrote the manuscript. Z-C.J. proposed and developed the concept, conceived and supervised the study, and revised the manuscript.

## Competing interests

The authors declare no competing interests.

## Additional information

Hong Liu or Zhi-Cheng Jing.

