## [Peer Review File · Nature Communications]

Single-cell analysis of peripheral blood from high-altitude pulmonary hypertension patients identifies a distinct monocyte phenotypeREVIEWER COMMENTS

Reviewer #1 (Remarks to the Author):

This study thoroughly reported immune cell atlas of peripheral blood mononuclear cell (PBMC) samples in high-altitude pulmonary hypertension (HAPH) patients using single-cell RNA sequencing (scRNA-seq). This study profiled the cellular composition and identified the prevalence of intermediate and nonclassical monocytes in PBMC samples from patients with HAPH than in those from the controls. The authors further propose that the marked changes in the monocyte phenotypes in all monocyte subsets, involves HIF-1 α downregulation and impaired gene expression in the phagocytosis, coagulation and wound healing pathways.

Comments

1. The authors have identified altered proportion of monocyte subsets from the bioinformatics analysis of scRNA-seq datasets (gene expression). Therefore, the authors should further validate those differential monocyte subsets identified by scRNA-seq in a small validation cohort using flow cytometric analysis? Flow cytometry can be performed using the bona fide markers to validate the frequency of immune subpopulations within the PBMC Fraction?

2. Particularly, did the find any significant correlation between the differential monocyte subsets or other PBMC cell types with any risk factors, age, sex (as in Morabito et al., Scand J Med Sci Sports. 2016, PMID: 26432186), echocardiographic parameters or general clinical measurements of HAPH patients? The authors should investigate and state the findings about any meaningful correlation of documented clinical parameters with scRNA-seq findings, which will be insightful.

3. Correlation between the clinical parameters versus differentially expressed genes (gene signatures (gene set) or individual expression of novel genes or known monocyte markers) should be analyzed and their association and relevance to hypoxia or HAPH should be discussed in appropriate sections.

4. The authors report that HIF-1 α was decreased in all HAPH samples compared to the control. Are the downstream target genes of HIF1 α also downregulated in C0 and C1 monocytes (HIF1 α transcriptional network)? Few representative plots of HIF1 α target genes can be added.

Reviewer #2 (Remarks to the Author):

In the manuscript "single cell RNA sequencing of peripheral blood from patients with high altitude pulmonary hypertension reveals a distinct monocyte phenotype", the authors have present new data from single cell RNA sequencing of peripheral blood mononuclear cells from cases and controls of high altitude residents with and without pulmonary hypertension. Herein, the authors were able to make a couple of very interesting observations from the rich gene expression data which may reveal some characteristics of the individuals with this disease but leave this reviewer curious about mechanism and physiology.

The first interesting finding that the authors highlight is that a subset of monocytes is upregulated in patients with high altitude pulmonary hypertension. These are termed 'intermediate monocytes', and are present in diseased individuals but virtually absent from controls. The induction of this subset of monocytes is sufficient to expand the total monocyte constituency in cases to be significantly greater than total monocytes in control subjects.

The second interesting finding that the authors highlight is that there is a relative deficiency in the gene product for HIF1 alpha expressed in monocyte cells from subjects with high altitude pulmonary hypertension.

Major comments:

-Given the nature of this disease and the population being studied, this report is mostly descriptive. Asking for further experimental data for a human disease like this would be impractical and probably unethical. However, without experimental exploration to follow up the findings presented, the extent to

which our understanding of the relevance of these observations to human disease will remain superficial.

-It is challenging to find robust physiologic signals from single cell gene expression data, and the authors do a respectable job in utilization of pathways analysis tools. However, whenever I see large numbers of pathways as being differential in cellular systems it always makes me question if I am really understanding the signal from the noise as it relates to real cellular behavior, in this case monocytes and HIF 1A. Also, for this figure the authors do not include cluster C0 in their comparisons, but the C0 cluster is the one that displays the largest difference in HIF 1A signaling and may have the most to do with the role of HIF 1A in the pathobiology of disease based on the rest of the manuscript.

-The findings on decreased levels of the HIF1A RNA in monocytes from affected individuals is intriguing. In cells, regulation of HIF 1A abundance by the ubiquitin proteasome considered to be principal in the activity of that protein. There is an opportunity here for the authors to use the gene expression information they have as a means to interrogate HIF 1A signaling. For example, evaluation of targeted gene products whose transcription is activated by HIF 1A (like EPO or VEGF) could indicate that there is a difference in HIF 1A activity between the groups.

I quite like the spirit of the analysis in the supplement showing communication maps. It is tempting to think that the cells in the blood would release factors that are active on cells outside those analyzed, particularly endothelial cells (which again makes me think that the evaluation of factors downstream of HIF 1A particularly growth factors may be of merit). I realize this may be impractical with the data at hand, but wonder if there could be a focused evaluation of DEGs from monocytes that could impact cells in the circulatory system.

Minor comments:

Collectively, we must be careful in our generalizations about gene expression data. Statements like the following are well intentioned but the data fall short of being proof for such a claim:

"...monocytes in HAPH patients showed significantly decreased levels of phagocytosis, myeloid cell differentiation, coagulation, platelet adhesion to exposed collagen, and regulation of TNF production (Figure 4D)."

The data collected are from gene expression not experiments which would actually indicate if these behaviors of cells were different between the groups.

"Previous reports have addressed the important role of HIFs in hypoxia-induced PH development (e.g., WHO group 3 PH) by using animal models" This statement needs citations

The manuscript will benefit from further editing for syntax and semantics. For example inline 436 the authors use 'distended' when I think 'disparate' may be more fitting.

Reviewer #3 (Remarks to the Author):

In the present study, the authors sorted human peripheral mononuclear cells from patients with high-altitude pulmonary hypertension (HAPH) and age matched controls and compared their immune profiles using single cell RNA sequencing. The authors found a significant expansion of monocytes in individuals with HAPH, primarily within the intermediate and non-classical subsets. Differential analysis indicates functional rewiring of monocyte subsets, and a substantial reduction of HIF1A transcripts with HAPH. While the design and methodology are straightforward, the major concern is that the manuscript is descriptive analysis of one large single cell experiment providing limited advancement and functional insight compared to what is already known in the field. If anything, this study validates previously reported role of non-classical monocytes in pulmonary hypertension in a murine model, published from the same lab. The study identifies a key marker of hypoxia downregulated with HAPH but fails to follow on its precise role in the pathophysiology of HAPH.

Major concerns:

1. The experimental design should have included patients with PH, to compare and contrast the

immune adaptations with HAPH.

2. While single cell analyses provide an unbiased snapshot of immune cell diversity and phenotypic changes, a more detailed analysis of cells using flow cytometry should accompany this study. Given that subsets within human PBMC are well characterized by flow, this should be fairly straightforward.
3. A more stringent fold change cutoff (than the current 0) should be adopted to weed out background noise from significant biological differences.
4. It is unclear why the authors chose to use a lung immune map to annotate human PBMC, the latter being fairly straightforward, given the plethora of single cell studies and catalogs of transcriptional markers for PBMC.
5. Cell to cell communication analysis too CellPhoneDB was designed identify ligand receptor interactions within tissue environment. The authors describe this in the methods but provide an inadequate interpretation of this analysis in the results. Instead, they should provide a bubble plot quantifying the interaction strength between receptors and ligands with significant differences in HAPH subjects.
6. It was very surprising to observe no DC (cDC and pDC) clusters in PBMC but a CD34+ HSC cluster, which are fairly rare in PBMC from aged subjects. Most likely the DC cluster has fused with other myeloid cells in the tSNE. The authors should use a different approach for clustering and/or annotation.
7. It is still unclear if non-classical monocytes play a role in the pathophysiology of HAPH in the blood or are merely responding to hypoxia. This needs to be discussed.
8. The order of controls and case samples in Fig 4D needs to be swapped. The differential analysis and functional enrichment of the DEGs suggests down-regulation of phagocytosis, coagulation, and regulation of cytokine (TNF) production but up-regulation of pathways associated with viral response. These findings should be functionally validated.
9. Finally, line 348 reads "data indicate that HAPH patients may suffer from severe HIF-1a deficiency". This interpretation is incorrect. The authors should either quantify HIF-1a protein in monocyte subsets or provide additional analyses on expression (down or up) of HIF-1a regulated genes in monocyte subsets.
10. Discussion lines 440-447 hints at HIF-1a's role in immune training. Is there evidence of immune training in monocytes from patients with PH or HAPH?

Minor Comments

1. Line 458 should read future "in vitro"
2. Line 463, the word "immune responses" should be replaced with "immune profiles".

Response to Reviewers

We are grateful for the thoughtful and helpful comments from all the reviewers and for the opportunity to revise our manuscript. In this extensively revised manuscript, we have performed substantial experiments as suggested by reviewers as much as possible. In addition, all supplementary data have been carefully reviewed and integrated to meet the requirement of the journal. Furthermore, we have carefully polished the whole manuscript to improve its readability and revised some inaccurate statements. Changes made in the revision have been highlighted in red. Please check our point-to-point response to each question or comments below.

Reviewer #1 (Remarks to the Author):

This study thoroughly reported immune cell atlas of peripheral blood mononuclear cell (PBMC) samples in high-altitude pulmonary hypertension (HAPH) patients using single-cell RNA sequencing (scRNA-seq). This study profiled the cellular composition and identified the prevalence of intermediate and nonclassical monocytes in PBMC samples from patients with HAPH than in those from the controls. The authors further propose that the marked changes in the monocyte phenotypes in all monocyte subsets, involves HIF-1a downregulation and impaired gene expression in the phagocytosis, coagulation and wound healing pathways.

Comments:

1. The authors have identified altered proportion of monocyte subsets from the bioinformatics analysis of scRNA-seq datasets (gene expression). Therefore, the authors should further validate those differential monocyte subsets identified by scRNA-seq in a small validation cohort using flow cytometric analysis? Flow cytometry can be performed using the bona fide markers to validate the frequency of immune subpopulations within the PBMC Fraction?

Response: Thank you for your professional and pertinent opinion. According to your suggestions, we have collected a validation cohort of 21 participants, including 11 HAPH patients and 10 controls to verify the immune subpopulations. Accordingly, the frequency of monocytes (CD68+ cells) and three monocyte subsets (C0: CD68+CD14+ cells; C1: CD68+CD14-CD16+ cells; C2: CD68+CD14-CD16- cells) in PBMCs from 11 HAPH patients and 10 controls were analyzed by flow cytometry

(Figure R1 or revised Figure 3h-i), with higher frequency of C1 and C2 in HAPH patients relative to that of controls. Moreover, the frequency of immune subpopulations including CD4+ T cells (CD3+CD4+ cells), CD8+ T cells (CD3+CD8+ cells), NK cells (CD3-CD16+CD56+ cells) within the PBMC fraction were also examined in Figure R2 or Figure S2c of revised supplementary materials. The cell fractions of the immune subpopulation in PBMCs from the validation cohort were shown in Table R1 or Table S6 of revised supplementary materials. Accordingly, no difference was detected in CD4+ T, CD8+ T or NK cells between HAPH and control subjects.

Figure R1. Frequency of monocyte clusters in PBMCs from patients with HAPH and controls. (A) Gating strategy of Monocytes by flow cytometry. (B) The comparison of the proportion of each monocyte subsets in PBMCs between HAPH patients (n=11) and controls (n=10) assessed by flow cytometry.

Figure R2. Frequency of main immune cell populations in PBMCs from patients with HAPH and controls. Gating strategy of T cells (CD4+ T and CD8+ T) and NK cells were displayed in upper

panel. The frequency of each population was quantified in lower panel.

Table R1. Characteristics and cell proportion in PBMCs from the validation cohort.

ID ₁	Group ₁	Gender ₁	Age ₁	Altitude (m) ₁	CD4/0.0% ₊		CD8/0.0% ₊		NK/0.0% ₊		
					CD3+CD4+ ₁	CD3+CD8+ ₁	CD3- CD16+CD56+ ₁	C0 ₁	C1 ₁	C2 ₁	
C6 ₁	Controls ₁	Female ₁	47 ₁	3194.97 ₁	30.8 ₁	25.1 ₁	17.7 ₁	49.3 ₁	23 ₁	7.62 ₁	↕
C7 ₁	Controls ₁	Female ₁	55 ₁	3194.97 ₁	32.6 ₁	16.2 ₁	13.1 ₁	45.7 ₁	30.9 ₁	8.61 ₁	↕
C8 ₁	Controls ₁	Male ₁	48 ₁	3194.97 ₁	40.3 ₁	15.5 ₁	5.73 ₁	68.1 ₁	32.9 ₁	9.42 ₁	↕
C9 ₁	Controls ₁	Male ₁	65 ₁	3194.97 ₁	40.6 ₁	11.4 ₁	8.17 ₁	61.9 ₁	26 ₁	5.09 ₁	↕
C10 ₁	Controls ₁	Female ₁	50 ₁	3194.97 ₁	33.1 ₁	18.1 ₁	15.3 ₁	64.9 ₁	37.9 ₁	7.21 ₁	↕
C11 ₁	Controls ₁	Female ₁	32 ₁	3109.72 ₁	13.9 ₁	30.7 ₁	16.6 ₁	55.5 ₁	17.2 ₁	11.8 ₁	↕
C12 ₁	Controls ₁	Female ₁	64 ₁	3194.97 ₁	32.6 ₁	22.7 ₁	6.52 ₁	81.5 ₁	12 ₁	1.7 ₁	↕
C13 ₁	Controls ₁	Female ₁	63 ₁	3109.72 ₁	38.3 ₁	16.1 ₁	16.5 ₁	68.3 ₁	33.1 ₁	9.39 ₁	↕
C14 ₁	Controls ₁	Male ₁	55 ₁	3194.97 ₁	34.6 ₁	21.8 ₁	33.9 ₁	50 ₁	5.7 ₁	4.54 ₁	↕
C15 ₁	Controls ₁	Female ₁	44 ₁	3194.97 ₁	36.7 ₁	19.8 ₁	15 ₁	68.9 ₁	12.1 ₁	3.35 ₁	↕
H1 ₁	Controls ₁	Female ₁	67 ₁	3194.97 ₁	29.7 ₁	25.7 ₁	7.86 ₁	73.9 ₁	30.9 ₁	6.92 ₁	↕
H2 ₁	HAPH ₁	Female ₁	76 ₁	3194.97 ₁	39.6 ₁	18.9 ₁	26.8 ₁	57.8 ₁	35 ₁	7.88 ₁	↕
H3 ₁	HAPH ₁	Female ₁	51 ₁	3194.97 ₁	38.2 ₁	20.5 ₁	34.1 ₁	50 ₁	47.2 ₁	19 ₁	↕
H4 ₁	HAPH ₁	Female ₁	60 ₁	3194.97 ₁	37.5 ₁	20 ₁	14.5 ₁	63.6 ₁	30.9 ₁	10.3 ₁	↕
H5 ₁	HAPH ₁	Male ₁	77 ₁	3194.97 ₁	27.4 ₁	25.4 ₁	27.8 ₁	59.6 ₁	38.4 ₁	9.21 ₁	↕
H6 ₁	HAPH ₁	Male ₁	72 ₁	3109.72 ₁	53 ₁	8.54 ₁	35.5 ₁	62.5 ₁	51 ₁	7.51 ₁	↕
H7 ₁	HAPH ₁	Female ₁	57 ₁	3194.97 ₁	38.7 ₁	16.3 ₁	23.5 ₁	50.1 ₁	27.7 ₁	17.9 ₁	↕
H8 ₁	HAPH ₁	Female ₁	48 ₁	3212.04 ₁	34.9 ₁	20 ₁	31.3 ₁	61.9 ₁	27.8 ₁	13.5 ₁	↕
H9 ₁	HAPH ₁	Female ₁	44 ₁	3212.04 ₁	39.7 ₁	15.1 ₁	18.7 ₁	38.6 ₁	47.4 ₁	13 ₁	↕
H10 ₁	HAPH ₁	Female ₁	56 ₁	3194.97 ₁	41.1 ₁	17.5 ₁	3.18 ₁	70.4 ₁	48.1 ₁	9.25 ₁	↕
H11 ₁	HAPH ₁	Female ₁	59 ₁	3194.97 ₁	38.6 ₁	19.3 ₁	9.15 ₁	48.7 ₁	18.5 ₁	11.7 ₁	↕

2. Particularly, did they find any significant correlation between the differential monocyte subsets or other PBMC cell types with any risk factors, age, sex (as in Morabito et al., Scand J Med Sci Sports. 2016, PMID: 26432186), echocardiographic parameters or general clinical measurements of HAPH patients? The authors should investigate and state the findings about any meaningful correlation of documented clinical parameters with scRNA-seq findings, which will be insightful.

Response: We have carefully investigated the correlation of documented clinical parameters and PBMC cell types and/or monocyte subsets using Pearson's correlation coefficient. According to the analysis, there were negative correlations between B cells and clinical measures including systolic blood pressure and aspartate amino transferase, between CD8+ T cells and age, and between NK cells and total protein. There were also positive correlations between CD8+ T cells and total protein, between memory B cells and white blood cell count, between monocyte and tricuspid valve max

regurgitation velocity (TVRVmax), between NK cells and hemoglobin, and between Tregs and pulse blood oxygen saturation. In terms of the correlation between clinical parameters and monocyte subsets, TVRVmax and systolic PAP, both were positively correlated with C0 monocyte or C2 monocyte. There was also a positive correlation between platelet count and C1 monocyte. However, the intrinsic relationship of the cell fractions and clinical parameter remains to be investigated in future studies. The correlation coefficient of clinical parameters and fractions of various PBMC cell types were shown in Table R2 or Table S6 of revised supplementary materials.

Table R2. The correlation coefficient of clinical parameters and fractions of various PBMC cell (sub) types

	B-cells	CD4+_ T-cells	CD8+_ T-cells	Memory_ B-cells	Monocytes	naive_ B-cells	NK_cells	Tregs	C0	C1	C2
Age	0.043	-0.235	-0.658 *	-0.245	0.284	-0.182	0.417	0.259	-0.126	0.011	0.081
Systolic blood pressure (SBP)	-0.649*	0.427	0.235	0.287	-0.494	-0.347	0.203	-0.144	0.088	0.322	-0.147
Diastolic blood pressure (DBP)	-0.341	0.098	-0.133	0.084	-0.112	-0.056	0.081	-0.295	-0.189	-0.021	0.084
Pulse blood oxygen saturation (SPO2)	-0.533	0.3	0.272	0.399	-0.293	-0.032	-0.081	-0.642*	0.187	-0.388	0.028
Body mass index (BMI)	-0.498	0.224	0.497	0.336	-0.364	-0.301	-0.021	-0.098	0.154	0.189	-0.126
White blood cell count (WBC)	-0.078	0.483	0.049	0.608*	-0.455	0.287	-0.007	-0.224	0.476	-0.531	-0.203
Red blood cell count (RBC)	0.438	0.14	-0.105	0.063	-0.329	0	0.448	0.126	0.51	0.175	-0.566
Hemoglobin (Hb)	-0.011	0.483	-0.49	-0.014	-0.333	-0.315	0.578*	0.256	0.298	0.102	-0.364
Platelet count (PLT)	-0.007	-0.014	0.301	0.028	0.021	0.21	-0.35	-0.434	0.336	-0.671*	0
Aspartate amino transferase (AST)	-0.67*	0.286	-0.229	-0.113	0.014	-0.497	0.018	-0.466	-0.236	-0.233	0.212
Total protein (TP)	-0.306	-0.392	0.818*	-0.371	0.322	-0.112	-0.678*	-0.483	-0.168	-0.182	0.273
Albumin (Alb)	0.219	-0.151	0.228	-0.273	0.032	-0.049	-0.375	-0.305	0.032	-0.193	-0.067
Total bilirubin (TBil)	0.064	0.294	-0.182	-0.252	0	0.098	0.21	0.077	0.133	0.133	-0.196
Uric acid (UA)	-0.264	0.203	-0.336	0.088	0.025	0.144	-0.095	0.049	-0.179	-0.333	0.214
Creatinine (CRE)	-0.326	0.269	-0.311	0.106	-0.325	-0.131	0.353	-0.272	0.424	-0.315	-0.258
Fasting blood-glucose (FBG)	0.488	-0.175	0.046	0.011	-0.025	0.245	-0.091	-0.06	0.315	-0.298	-0.249
Pulmonary artery dimension (PAD)	-0.396	0.011	0.014	0.125	0.125	0.136	-0.132	-0.425	-0.411	-0.154	0.439
Left ventricular ejection fraction (LVEF)	0.066	0.458	0.194	0.285	-0.246	0.183	-0.148	0.204	-0.243	0.313	0.046
Right atrium diameter (RAD)	-0.248	-0.474	0.148	-0.357	0.474	-0.216	-0.382	-0.046	-0.421	-0.219	0.445
Right ventricle diameter (RVD)	-0.223	0.186	-0.189	0.133	-0.126	0.105	-0.147	0.06	-0.077	-0.203	0.056
Tricuspid valve max Regurgitation Velocity (m/s) (TVRVmax)	-0.508	-0.406	0.032	-0.399	0.596*	-0.233	-0.353	-0.243	-0.617*	-0.317	0.723*
Pulmonary systolic pressure (PSP)	-0.518	-0.379	0.053	-0.358	0.561	-0.182	-0.365	-0.242	-0.586*	-0.326	0.702*

3. Correlation between the clinical parameters versus differentially expressed genes (gene signatures (gene set) or individual expression of novel genes or known monocyte markers) should be analyzed and their association and relevance to hypoxia or HAPH should be discussed in appropriate sections.

Response: We have performed the spearman correlation analysis between clinical parameters versus genes including *HIF-1A* and its target gene *VEGFA*. The correlation coefficient of clinical parameters and gene expression were shown in **Table R3** or **Table S8** of revised supplementary materials. According to the analysis, there was a strong positive correlation between red blood cell count and VEGFA, and a weak positive correlation between creatine and HIF1A as well as between fasting blood glucose and VEGFA.

Table R3. The correlation and *p*-value of clinical parameters and expression of HIF1A and its target gene VEGFA.

Clinical parameters	HIF1A		VEGFA	
	Correlation	Pvalue	Correlation	Pvalue
Age	0.172	0.594	0.042	0.897
Systolic blood pressure (SBP)	-0.525	0.079	-0.252	0.429
Diastolic blood pressure (DBP)	0.095	0.770	-0.081	0.803
Pulse blood oxygen saturation (SPO2)	-0.222	0.488	-0.247	0.439
Body mass index (BMI)	-0.420	0.175	-0.217	0.499
White blood cell count (WBC)	-0.056	0.863	-0.105	0.746
Red blood cell count (RBC)	0.385	0.217	0.720 *	0.008
Hemoglobin (Hb)	0.067	0.837	0.238	0.456
Platelet count (PLT)	0.175	0.587	-0.021	0.948
Aspartate amino transferase (AST)	0.028	0.931	-0.275	0.387
Total protein (TP)	0.035	0.914	-0.140	0.665
Albumin (Alb)	0.483	0.111	0.235	0.463
Total bilirubin (Tobol)	0.210	0.513	0.413	0.183
Uric acid (UA)	0.039	0.905	-0.382	0.221
Creatinine (CRE)	0.081 *	0.035	0.913	0.802
Fasting blood- glucose (FBG)	0.620	0.466	0.127 *	0.032
Pulmonary artery dimension (PAD)	-0.322	0.443	0.149	0.308
Left ventricular ejection fraction (LVEF)	-0.303	0.141	0.662	0.339
Right atrium diameter (RAD)	0.053	0.396	0.203	0.870
Right ventricle diameter (RVD)	0.119	0.273	0.390	0.712
Tricuspid valve max Regurgitation Velocity (m/s) (TVRVmax)	-0.325	0.737	0.006	0.303
Pulmonary systolic pressure (PSP)	-0.326	0.737	0.006	0.301

4. The authors report that HIF-1 α was decreased in all HAPH samples compared to the control. Are the downstream target genes of HIF1a also downregulated in C0 and C1 monocytes (HIF1a transcriptional network)? Few representative plots of HIF1a target genes can be added.

Response: We appreciate your suggestions. We first obtained target genes of HIF-1A using BIND, HPRD and/or BioGRID. Then we performed differentially expressed gene analysis in C0, C1 and C2, in comparison of case and control group. It turned out that the expression of VEGFA in HAPH patients was much lower independent of monocyte subsets in comparison of control subjects, a similar trend as that of HIF-1A between the two groups. The representative boxplot of target gene *VEGFA* was added in **Figure R3** or **Figure 6** in our revised manuscript.

Figure R3. Comparison of the expression of HIF-1 α and VEGF1 in immune cell population between HAPH and controls. (g-i) Box plots showing the comparison of the expression of VEGFA in each monocyte cluster between HAPH patients (n=7) and controls (n=5).

Reviewer #2 (Remarks to the Author):

In the manuscript “single cell RNA sequencing of peripheral blood from patients with high altitude pulmonary hypertension reveals a distinct monocyte phenotype”, the authors have present new data from single cell RNA sequencing of peripheral blood mononuclear cells from cases and controls of high altitude residents with and without pulmonary hypertension. Herein, the authors were able to make a couple of very interesting observations from the rich gene expression data which may reveal some characteristics of the individuals with this disease but leave this reviewer curious about mechanism and physiology.

The first interesting finding that the authors highlight is that a subset of monocytes is upregulated in patients with high altitude pulmonary hypertension. These are termed ‘intermediate monocytes’, and are present in diseased individuals but virtually absent from controls. The induction of this subset of monocytes is sufficient to expand the total monocyte constituency in cases to be

significantly greater than total monocytes in control subjects.

The second interesting finding that the authors highlight is that there is a relative deficiency in the gene product for HIF1 alpha expressed in monocyte cells from subjects with high altitude pulmonary hypertension.

Major comments:

-Given the nature of this disease and the population being studied, this report is mostly descriptive. Asking for further experimental data for a human disease like this would be impractical and probably unethical. However, without experimental exploration to follow up the findings presented, the extent to which our understanding of the relevance of these observations to human disease will remain superficial.

Response: Thank you for your professional and pertinent opinion. According to your suggestions, we have collected a validation cohort of 21 participants, including 11 HAPH patients and 10 controls to verify the immune subpopulations. Accordingly, the frequency of monocytes (CD68+ cells) and three monocyte subsets (C0: CD68+CD14+ cells; C1: CD68+CD14-CD16+ cells; C2: CD68+CD14-CD16- cells) in PBMCs from 11 HAPH patients and 10 controls were analyzed by flow cytometry (**Figure R4 or revised Figure 3h-i**), with higher frequency of C1 and C2 in HAPH patients relative to that of controls. Moreover, the frequency of immune subpopulations including CD4+ T cells (CD3+CD4+ cells), CD8+ T cells (CD3+CD8+ cells), NK cells (CD3-CD16+CD56+ cells) within the PBMC fraction were also examined in **Figure R5 or revised Figure S2c**. The cell fractions of the immune subpopulation in PBMCs from the validation cohort were shown in revised **Table S6**. Accordingly, no difference was detected in CD4+ T, CD8+ T or NK cells between HAPH and control subjects.

Figure R4. Frequency of monocyte clusters in PBMCs from patients with HAPH and controls. (A) Gating strategy of Monocytes by flow cytometry. (B) The comparison of the proportion of each monocyte subsets in PBMCs between HAPH patients (n=11) and controls (n=10) assessed by flow cytometry.

Figure R5. Frequency of of main immune cell populations in PBMCs from patients with HAPH and controls. Gating strategy of T cells (CD4+ T and CD8+ T) and NK cells were displayed in upper panel. The frequency of each population was quantified in lower panel.

As you mentioned, further study of the mechanism in patients is not advisable. To this end, we established an animal model of PH to evaluate the function of the candidate genes derived from single-cell sequencing of patients with HAPH.

First, we determined the establishment of the hypoxia-induced PH mouse model by right heart catheterization, right heart hypertrophy index, and histopathological evaluation. We verified the mRNA levels of the candidate genes in the lung tissues of hypoxic PH mice, and further verified them at the protein level. Next, we used an *in vitro* co-culture model of PASMCs and Jurkat Clone E6-1 cells to further analyze the roles of genes on the cellular phenotype. Taken together, we confirmed that CD28 plays a role in regulating disease progression in PH. Please check the data in

details below (Figure R6), which were also added into the Figure 5 or Supplementary Materials (Figure S8-11 and Table S1-2) in our revised manuscript.

Figure R6. Gene-function validation in hypoxic PH mice. (a,b) C57BL/6 mice exposed to hypoxia (10% O₂) for 4 weeks exhibited higher RVSP (a) and RVHI (b) compared to that of mice in normoxia (21% O₂) (n = 10 for per group, ****p* < 0.001; Student's *t*-test. Data were presented as mean ± SD). (c) Representative images of hematoxylin and eosin (H&E) staining of lung tissues from mice under normoxic or hypoxic condition. Scale bars, 20 μm. (d) Assessment of pulmonary vascular remodeling by determination of the ratio of the media cross-sectional area to the total vessel cross-sectional area in each group (n = 10 for per group, ****p* < 0.001; Student's *t*-test. Data were presented as mean ± SD). (e) The expression of *Prf1*, *Spon2*, *Tbx21*, *Cd28* and *Cd3e* in lung tissues of the mice under normoxic or hypoxic condition at mRNA levels (n = 7-8 for per group, **p* < 0.05, ***p* < 0.01; Student's *t*-test. Data were presented as mean ± SD). (f-i) Representative images of immunoblottings and the quantification of the expression of *Cd28* (g), *Spon2* (h) and *Tbx21* (i) in lung tissues of the control mice and hypoxic PH mice at protein levels (n = 8 for per group, ***p* < 0.01, ****p* < 0.001; Student's *t*-test. Data were presented as mean ± SD). (j) Representative images of immunoblottings of Jurkat Clone E6-1 cells transfected with CD28 siRNA (siCD28) or control siRNA (siControl). (k) Viability of human PASMCs co-cultured with or without CD28-silencing or control Jurkat cells for 48 h (n = 5 for per group, **p* < 0.05, ***p* < 0.01; one-way ANOVA, Tukey's *post-hoc* test. Data were presented as mean ± SD).

-It is challenging to find robust physiologic signals from single cell gene expression data, and the authors do a respectable job in utilization of pathways analysis tools. However, whenever I see large numbers of pathways as being differential in cellular systems it always makes me question if I am really understanding the signal from the noise as it relates to real cellular behavior, in this case monocytes and HIF 1A. Also, for this figure the authors do not include cluster C0 in their comparisons, but the C0 cluster is the one that displays the largest difference in HIF 1A signaling and may have the most to do with the role of HIF 1A in the pathobiology of disease based on the rest of the manuscript.

Response: As you mentioned the signal from the noise is sometimes related to real cellular behavior, we then determined to establish a hypoxia-induced PH mouse model to evaluate the function of the candidate genes derived from single-cell RNA sequencing distinguishing patients with high altitude PH from controls. In addition, we have added C0 cluster in **Figure 4d (or Figure R7 below)** for the comparison of pathway signature score in our revised manuscript. We found that HAPH patients had lower signature scores relevant to the pathways (eg. phagocytosis, myeloid cell differentiation, coagulation, platelet adhesion to exposed collagen, and regulation of TNF production etc.) compared to that of controls independent of monocyte subsets.

Figure R7. Differential biological pathways in each monocyte cluster between HAPH patients and controls. (d) Boxplot showing the mean pathway signature score of each monocyte subset from each group. *** $P < 0.001$; **** $P < 0.0001$; using unpaired Wilcox rank sum test.

-The findings on decreased levels of the HIF1A RNA in monocytes from affected individuals is intriguing. In cells, regulation of HIF 1A abundance by the ubiquitin proteasome considered to be principal in the activity of that protein. There is an opportunity here for the authors to use the gene expression information they have as a means to interrogate HIF 1A signaling. For example, evaluation of targeted gene products whose transcription is activated by HIF 1A (like EPO or VEGF) could indicate that there is a difference in HIF 1A activity between the groups.

Response: According to your advice, we focused on *HIF1a* target gene (VEGF) to investigate the difference in *HIF 1a* activity between HAPH patients and controls. It was shown that VEGF was downregulated in each subcluster of monocytes in PBMCs from HAPH relative to that of controls. The representative boxplots of target gene *VEGFA* were added in **Figure 6g-i** (or **Figure R8** below) as suggested in our revised manuscript.

Figure R8. Comparison of the expression of HIF-1 α and VEGF1 in immune cell population between HAPH and controls. (g-i) Box plots showing the comparison of the expression of VEGFA in each monocyte cluster between HAPH patients (n=7) and controls (n=5).

-I quite like the spirit of the analysis in the supplement showing communication maps. It is tempting to think that the cells in the blood would release factors that are active on cells outside those analyzed, particularly endothelial cells (which again makes me think that the evaluation of factors downstream of HIF 1A particularly growth factors may be of merit). I realize this may be impractical with the data at hand, but wonder if there could be a focused evaluation of DEGs from monocytes that could impact cells in the circulatory system.

Response: According to the previous cell-cell communication maps, we further focused on the top ligand and receptor pairs referring to monocytes C1/C2 with other cell types in case and control

group, respectively (Figure R9 or Figure S6G of revised manuscript; Figure R10 or Figure S7G of revised manuscript).

Figure R9. Cell-cell communication networks between C1 monocyte and other peripheral blood cell types using CellPhoneDB in (A-C) and control (D-E) individuals. Top 20 ligand-receptor pairs in case and control group (G).

G

Figure R10. Cell-cell communication networks between C2 monocyte and other peripheral blood cell types using CellPhoneDB in HAPH (A-C) and control (D-E) individuals. Top 20 ligand-receptor pairs in case and control group (G).

Minor comments:

Collectively, we must be careful in our generalizations about gene expression data. Statements like the following are well intentioned but the data fall short of being proof for such a claim:

“...monocytes in HAPH patients showed significantly decreased levels of phagocytosis, myeloid cell differentiation, coagulation, platelet adhesion to exposed collagen, and regulation of TNF production (Figure 4D).”

The data collected are from gene expression not experiments which would actually indicate if these behaviors of cells were different between the groups.

Response: We apologize for the improper statement and revised the sentence as below:

“We found that HAPH patients had lower signature scores relevant to the pathways (eg. phagocytosis, myeloid cell differentiation, coagulation, platelet adhesion to exposed collagen, and regulation of TNF production etc.) compared to that of controls independent of monocyte subsets, while more obviously in C1 (non-classical) and C2 (intermediate) monocytes.”. In order to examine the role of genes in the process of PAH, we ranked the contributions of genes involved in each biological pathway in the **Figure 4d**, and selected 3 genes with the largest fold change in each biological pathway. Next, we established hypoxia-induced PH mouse model and found that Cd28, Tbx21, and Spon2 were also significantly elevated in the lung tissues of hypoxic PH mice (please see **Figure R6** above or **Figure 5f-5i, Figure S9** of our revised manuscript), in line with their transcriptional alterations.

“Previous reports have addressed the important role of HIFs in hypoxia-induced PH development (e.g., WHO group 3 PH) by using animal models” This statement needs citations.

Response: We have added the necessary references (PMID: 31515405; PMID: 32881714) in our revised manuscript.

The manuscript will benefit from further editing for syntax and semantics. For example inline 436 the authors use ‘distended’ when I think ‘disparate’ may be more fitting.

Response: We have edited and polished our manuscript accordingly and revised the description as suggested in the revised version.

Reviewer #3 (Remarks to the Author):

In the present study, the authors sorted human peripheral mononuclear cells from patients with high-altitude pulmonary hypertension (HAPH) and age matched controls and compared their immune profiles using single cell RNA sequencing. The authors found a significant expansion of monocytes in individuals with HAPH, primarily within the intermediate and non-classical subsets. Differential analysis indicates functional rewiring of monocyte subsets, and a substantial reduction of HIF1A transcripts with HAPH. While the design and methodology are straightforward, the major concern is that the manuscript is descriptive analysis of one large single cell experiment providing limited advancement and functional insight compared to what is already known in the field. If anything, this study validates previously reported role of non-classical monocytes in pulmonary hypertension in a murine model, published from the same lab. The study identifies a key marker of hypoxia downregulated with HAPH but fails to follow on its precise role in the pathophysiology of HAPH.

Major concerns:

1. The experimental design should have included patients with PH, to compare and contrast the immune adaptations with HAPH.

Response: Thank you for your comments. Accordingly, we have collected PBMCs from 6 patients with PH and performed single cell RNA sequencing. The PH immune landscape was depicted following the same analysis workflow. We identified 10 cell types according to the canonical markers, including monocytes, NK cells, CD4+, CD8+, and regulatory T-cells, memory and naive B-cells, cDC, pDC and HSCs (**Figure R11** or **Figure 7a-c** of revised manuscript). Among them, monocytes accounted for 26.7%. The landscape comparison of HAPH and PH showed that their immune adaptations were similar, both in terms of cell type composition and cell fractions in each population (**Figure R11** or **Figure 7d-e** of revised manuscript). FACS analysis also indicated similar cell proportions of CD4+ T-cells, CD8+ T-cells and NK cells in PH and HAPH (**Figure R11** or **Figure 7f-g** of revised manuscript). The PH immune landscape was depicted following the same analysis workflow. We identified 10 cell types according to the canonical markers, including monocytes, NK cells, CD4+, CD8+, and regulatory T-cells, memory and naive B-cells, cDC, pDC and HSCs (**Figure 7a-c**). Among them, monocytes accounted for 26.7%. The landscape comparison of HAPH and PH showed that their immune adaptations were similar, both in terms of cell type composition and cell fractions in each population (**Figure 7d-e**). FACS analysis also indicated

similar cell proportions of CD4+ T-cells, CD8+ T-cells and NK cells in PH and HAPH (Figure R11 or Figure 7f-g).

Figure R11. Immune cell composition in the circulation system of patients with PH and its comparison with those in HAPH. (a) t-SNE plot of the main immune cell subsets in PBMCs from patients with PH (n=6). (b) Dot plot depicting the percentages and average expressions of the canonical genes associated with each main immune cell cluster in PBMCs from patients with PH (n=6). (c) Violin plots indicating the gene expressions in each main cluster of cells from PH patients

(n=6). **(d)** Boxplots comparing the percentages of each main cell type in PBMCs between HAPH (n=7) and PH (n=6) patients. The two-sided p values from the Wilcoxon rank-sum test were shown. **(e)** Proportions of each cell type in each sample as indicated. **(f-h)** The proportion of T cells, NK cells and C0, C1, C2 monocyte subsets in PBMCs from HAPH (n=11) and PH (n=6) patients as analyzed by flow cytometry ($*p < 0.05$; Student's t -test or Mann-Whitney test as appropriate. Data were presented as mean \pm SD).

2. While single cell analyses provide an unbiased snapshot of immune cell diversity and phenotypic changes, a more detailed analysis of cells using flow cytometry should accompany this study. Given that subsets within human PBMC are well characterized by flow, this should be fairly straightforward.

Response: Accordingly, the frequency of monocytes (CD68+ cells) and three monocyte subsets (C0: CD68+CD14+ cells; C1: CD68+CD14-CD16+ cells; C2: CD68+CD14-CD16- cells) in PBMCs from 11 HAPH patients and 10 controls were analyzed by flow cytometry (**Figure R12 or revised Figure 3h-i**), with higher frequency of C1 and C2 in HAPH patients relative to that of controls. Moreover, the frequency of immune subpopulations including CD4+ T cells (CD3+CD4+ cells), CD8+ T cells (CD3+CD8+ cells), NK cells (CD3-CD16+CD56+ cells) within the PBMC fraction were also examined in **Figure R13 or revised Figure S2c**. The cell fractions of the immune subpopulation in PBMCs from the validation cohort were shown in revised **Table S6**. Accordingly, no difference was detected in CD4+ T, CD8+ T or NK cells between HAPH and control subjects.

Figure R12. Frequency of monocyte clusters in PBMCs from patients with HAPH and controls. **(A)** Gating strategy of Monocytes by flow cytometry. **(B)** The comparison of the proportion of each monocyte subsets in PBMCs between HAPH patients (n=11) and controls (n=10) assessed by flow cytometry.

Figure R13. Frequency of of main immune cell populations in PBMCs from patients with HAPH and controls. Gating strategy of T cells (CD4+ T and CD8+ T) and NK cells were displayed in upper panel. The frequency of each population was quantified in lower panel.

3. A more stringent fold change cutoff (than the current 0) should be adopted to weed out background noise from significant biological differences.

Response: Actually, we output all the detected genes, as well as the fold change. The differentially expressed gene HIF1A met the criteria of fold change > 2 in comparison of case and control group in monocytes. We have revised the description in our revised manuscript.

4. It is unclear why the authors chose to use a lung immune map to annotate human PBMC, the latter being fairly straightforward, given the plethora of single cell studies and catalogs of transcriptional markers for PBMC.

Response: We are sorry for the confused description in the method section. We have revised the description as below and also in our revised manuscript where we wrote:

“Cell type annotations were performed on the Blueprint and Encode reference dataset via SingleR, which assigns cellular identity for single cell transcriptomes by comparison to reference datasets of

pure cell types sequenced by microarray or RNA-seq. In detail, we corrected the annotations of SingleR according to the following: (i) “Class-switched_mem”, “Class-switched memory B-cells” and “Memory_B-cells” were defined as “Memory_B-cells”; (ii) “CD8+_T-cells”, “CD8+_Tcm” and “CD8+_Tem” were defined as “CD8+_T-cells”; and (iii) “CD4+_T-cells”, “CD4+_Tcm” and “CD4+_Tem” were defined as “CD4+_T-cells”.

5. Cell to cell communication analysis too CellPhoneDB was designed identify ligand receptor interactions within tissue environment. The authors describe this in the methods but provide an inadequate interpretation of this analysis in the results. Instead, they should provide a bubble plot quantifying the interaction strength between receptors and ligands with significant differences in HAPH subjects.

Response: Accordingly, we have provided a bubble plot quantifying the interaction strength between receptors and ligands with significant differences in HAPH subjects (**Figure R14** or **Figure S6G** of revised manuscript; **Figure R15** or **Figure S7G** of revised manuscript).

Figure R14. Cell-cell communication networks between C1 monocyte and other peripheral blood cell types using CellPhoneDB in (A-C) and control (D-E) individuals. Top 20 ligand-receptor pairs in case and control group (G).

G

Figure R15. Cell-cell communication networks between C2 monocyte and other peripheral blood cell types using CellPhoneDB in HAPH (A-C) and control (D-E) individuals. Top 20 ligand-receptor pairs in case and control group (G).

6. It was very surprising to observe no DC (cDC and pDC) clusters in PBMC but a CD34+ HSC

cluster, which are fairly rare in PBMC from aged subjects. Most likely the DC cluster has fused with other myeloid cells in the tSNE. The authors should use a different approach for clustering and/or annotation.

Response: Thank you for your professional comments. As suggested, we revised the annotation and found a cluster of cDC characterized by the classical marker gene CD1C. The corresponding result was shown in **Figure R16** or **Figure S12** of revised manuscript.

Figure R16. Differential immune cell composition in the periphery of patients with HAPH. (a)

tSNE (t-distributed stochastic neighbor embedding) plot of the main immune cell subsets. **(b)** Dot plot depicting the percentages and average expressions of the canonical genes associated with each main immune cell cluster. **(c)** Violin plots indicating the gene expressions in each main cluster (cluster numbers as in Figure 2A) of cells from both the HAPH and control PBMC samples. **(d)** Boxplots comparing the percentages of each main cell type between the HAPH (n=7) and control (n=5) PBMC samples. The x axes correspond to each group. The two-sided P values from the Wilcoxon rank-sum test are shown. **(e)**. Proportions of each cell type in each sample.

7. It is still unclear if non-classical monocytes play a role in the pathophysiology of HAPH in the blood or are merely responding to hypoxia. This needs to be discussed.

Response: As suggested, we investigated literatures and discussed the roles of non-classical monocytes as follows.

Non-classical monocyte (including intermediate and non-classical) were regarded as an important component of monocyte with an expression of CD16. It mainly patrols in the peripheral blood. Non-classical monocyte is related to immune and inflammatory responses, and has been reported to be elevated in pathological conditions such as tumor and sepsis. In cardiovascular diseases, non-classical monocyte prevailed in patients with coronary heart disease and stroke. In the case of obstructive sleep apnea syndrome, the proliferation of intermediate and non-classical CD16+ monocyte were observed, indicative of a correlation to hypoxia (PMID: 33268482). However, the role of non-classical monocyte in high altitude pulmonary hypertension has not been reported to date. It has been reported that CD16+ monocytes were associated with systematic sclerosis and involved in the pathogenesis of pulmonary arterial vasculopathy. Statin treatment prevents the development of pulmonary arterial hypertension in a nonhuman primate model of HIV-associated PAH, accompanying with decreased ratio of CD16+ monocytes (PMID: 31882598). Combined with our results, we postulated that the elevation of non-classical monocyte in patients with high altitude pulmonary hypertension is associated with hypoxia and pathophysiology of HAPH. However, the intrinsic correlation and its precise mechanism underlying pulmonary hypertension are worthy of further studies.

8. The order of controls and case samples in Fig 4D needs to be swapped. The differential analysis and functional enrichment of the DEGs suggests down-regulation of phagocytosis, coagulation, and

regulation of cytokine (TNF) production but up-regulation of pathways associated with viral response. These findings should be functionally validated.

Response: The order of controls and case samples in **Figure 4d** has been swapped in the revised manuscript according to your comments.

In order to examine the role of genes in the process of PH, we ranked the contributions of genes involved in each biological pathway in the Figure 4d, and selected 3 genes with the largest fold change in each biological pathway. A total of 15 candidate genes were determined from the 7 pathways. Firstly, we established a hypoxia-induced PH model in mice, with higher RVSP identified by right heart catheterization (**Figure R17** or **Figure 5a** of revised manuscript), right ventricular hypertrophy index (**Figure R17** or **Figure 5b** of revised manuscript), and histopathological evaluation (**Figure R17** or **Figure 5c-d** of revised manuscript). We analyzed the mRNA expression levels of 15 genes in the lung tissue of hypoxic PH mice, and found that the expression of 5 genes (*Prf1*, *Spon2*, *Tbx21*, *Cd28* and *Cd3e*) were significantly increased compared to that in control mice (**Figure R17** or **Figure 5e** of revised manuscript), which were consistent with the results of ScRNA sequencing in HAPH patients. The rest 10 genes showed no significant difference in lung tissue of hypoxic PH mice (**Figure S8**). Next, we quantitatively analyzed 5 genes at the protein level, and found that Cd28, Tbx21, and Spon2 were significantly elevated in the lung tissues of hypoxic PH mice (**Figure R17** or **Figure 5f-i**, **Figure S9** of revised manuscript), in line with their transcriptional levels. The expression levels of Spon2 and Prf1 were not effectively detected in the lung tissues of hypoxic PH mice (**Figure S9** of revised manuscript). Co-cultured with Jurkat Clone E6-1 cells induced an increase in the cell viability of human PSMCs, and the knockdown of *CD28* in Jurkat Clone E6-1 cells significantly inhibited this effect (**Figure R17** or **Figure 5j-k** of revised manuscript). Neither *TBX21* nor *SPON2* knockdown exhibited similar significant inhibitory effects (**Figure R17** or **Figure S10-11** of revised manuscript). All the above details have been added to the revised manuscript.

Figure R17. Gene-function validation in hypoxic PH mice. (a,b) C57BL/6 mice exposed to hypoxia (10% O₂) for 4 weeks exhibited higher RVSP (a) and RVHI (b) compared to that of mice in normoxia (21% O₂) (n = 10 for per group, ****p* < 0.001; Student's *t*-test. Data were presented as mean ± SD). (c) Representative images of hematoxylin and eosin (H&E) staining of lung tissues from mice under normoxic or hypoxic condition. Scale bars, 20 μm. (d) Assessment of pulmonary vascular remodeling by determination of the ratio of the media cross-sectional area to the total vessel cross-sectional area in each group (n = 10 for per group, ****p* < 0.001; Student's *t*-test. Data were presented as mean ± SD). (e) The expression of *Prf1*, *Spon2*, *Tbx21*, *Cd28* and *Cd3e* in lung tissues of the mice under normoxic or hypoxic condition at mRNA levels (n = 7-8 for per group, **p* < 0.05, ***p* < 0.01; Student's *t*-test. Data were presented as mean ± SD). (f-i) Representative images of immunoblottings and the quantification of the expression of Cd28 (g), Spon2 (h) and Tbx21 (i) in lung tissues of the control mice and hypoxic PH mice at protein levels (n = 8 for per group, ***p* < 0.01, ****p* < 0.001; Student's *t*-test. Data were presented as mean ± SD). (j) Representative images of immunoblottings of Jurkat Clone E6-1 cells transfected with CD28 siRNA (siCD28) or control siRNA (siControl). (k) Viability of human PASMCs co-cultured with or without CD28-silencing or control Jurkat cells for 48 h (n = 5 for per group, **p* < 0.05, ***p* < 0.01; one-way ANOVA, Tukey's *post-hoc* test. Data were presented as mean ± SD).

9. Finally, line 348 reads “data indicate that HAPH patients may suffer from severe HIF-1a deficiency”. This interpretation is incorrect. The authors should either quantify HIF-1a protein in

monocyte subsets or provide additional analyses on expression (down or up) of HIF-1 α regulated genes in monocyte subsets.

Response: Thank you for your professional and pertinent opinion. As suggested, we revised the sentence as below:

“our data indicate that HAPH patients may suffer from impairment of HIF-1 α -mediated signaling in the monocytes of the circulating system”

In addition, we obtained target genes of HIF-1 α using BIND, HPRD and/or BioGIRD. Then we performed differential expressed gene analysis in C0, C1 and C2, in comparison of case and control group. The representative boxplot of target gene VEGFA was added in **Figure 6g-i** of revised manuscript (please see **Figure R18** below).

Figure R18. Comparison of the expression of HIF-1 α and VEGF1 in immune cell population between HAPH and controls. (g-i) Box plots showing the comparison of the expression of VEGFA in each monocyte cluster between HAPH patients (n=7) and controls (n=5).

10. Discussion lines 440-447 hints at HIF-1 α 's role in immune training. Is there evidence of immune training in monocytes from patients with PH or HAPH?

Response: Thank you for your careful examination. The overstatement had been deleted in the revised manuscript.

Minor Comments

Line 458 should read future “in vitro”

Response: We have corrected it as suggested.

Line 463, the word “immune responses” should be replaced with “immune profiles”.

Response: We have revised it as suggested.

REVIEWER COMMENTS

Reviewer #1 (Remarks to the Author):

The authors have addressed all the comments.

Reviewer #2 (Remarks to the Author):

I submitted a more comprehensive critique and review on the initial submission. The authors have submitted a great deal of supporting material and I feel the work is now very well presented. The manuscript has been significantly edited and improved. I have no further requests from the authors regarding data.

Reviewer #3 (Remarks to the Author):

This is a revised manuscript that examined the impact altitude mediated pulmonary hypertension (HAPH) on the transcriptional landscape of monocytes. Despite the revisions provided by the authors, significant concerns remain with data analysis and interpretation. The major concerns are listed below:

1. Analysis:

- a. The authors conducted single cell RNA Sequencing on several HAPH, PH and healthy controls. A single UMAP of all the samples should have been generated to identify clusters within the same analysis parameters and multiple group comparisons of frequencies of the major subsets conducted together rather than being spread over multiple figures
- b. Differential gene expression still lists log₂FC of 0 – which is most likely the reason for identification of Tcell genes CD3, CD28, PRF1 as being differentially expressed between monocyte subsets.
- c. The DEG should be divided into up and down-regulated genes and enriched separately
- d. The authors do not carry out any DEG analysis in other subsets. A lack of difference in frequency does not equate a lack of transcriptional changes. This is further highlighted by the follow up experiments that utilized a T cell line and endothelial cells for follow up experiments
- e. No functional assays were carried out to interrogate the differences in module scores and transcriptional profile

2. Validation

- a. The experiments in mouse model use total lung tissue and do not specifically target monocytes/macrophages
- b. There is no validation for the reduced module score for HIF-1a signaling. There are antibodies that can be used to examine HIF-1a protein expression and monocyte/macrophage cell lines that can be used if primary cells are not an option.

3. Data visualization minor concerns:

- a. bubble plots and violin plots of marker genes are redundant and only one should be shown.
- b. Individual samples should be put in the supplemental figures.
- c. Feature plots should also be provided as supplemental figures rather than main figure – rather a main UMAP with clusters identified should be provided
- d. Heat map of marker genes for monocyte subsets without any gene names is not informative – there is no reason to show figure 2C- this more complete list can be provided as a supplementary table
- e. Gating strategy for monocytes is highly unusual – it should be a CD14 versus CD16 dot plot

Point-by-point response to the reviewers' comments

REVIEWER COMMENTS

Reviewer #1 (Remarks to the Author):

The authors have addressed all the comments.

Reviewer #2 (Remarks to the Author):

I submitted a more comprehensive critique and review on the initial submission. The authors have submitted a great deal of supporting material and I feel the work is now very well presented. The manuscript has been significantly edited and improved. I have no further requests from the authors regarding data.

Reviewer #3 (Remarks to the Author):

This is a revised manuscript that examined the impact altitude mediated pulmonary hypertension (HAPH) on the transcriptional landscape of monocytes. Despite the revisions provided by the authors, significant concerns remain with data analysis and interpretation. The major concerns are listed below:

1. Analysis:

a. The authors conducted single cell RNA Sequencing on several HAPH, PH and healthy controls. A single UMAP of all the samples should have been generated to identify clusters within the same analysis parameters and multiple group comparisons of frequencies of the major subsets conducted together rather than being spread over multiple figures.

Response: According to your suggestion, we generated a single tSNE and/or UMAP of all the samples to identify clusters within the same analysis parameters and multiple group comparisons of frequencies of the major subsets. Please see details in **Figure R1-2** or **Figure S14-15** of revised supplementary materials.

Figure R1. Differential immune cell composition in the circulation of patients with HAPH or PH. (a,b) t-SNE (t-distributed stochastic neighbor embedding) plot of the main immune cell subsets, color-coded for three groups (a, left) or nine clusters (a, right), or separate three groups (b). (c) Boxplots comparing the percentages of indicated cell types in PBMCs

between HAPH patients (n=7), PH patients (n=6) and control subjects (n=5). The two-sided p values from the Wilcoxon rank-sum test were shown.

Figure R2. Monocyte clusters in PBMCs from patients with HAPH, PH and controls. (a) Boxplots comparing the percentages of each monocyte cluster in PBMCs between HAPH patients (n=7), PH patients (n=6) and control subjects (n=5). The two-sided p values from the Wilcoxon rank-sum test were shown. (b) Proportions of each cell type in HAPH patients (n=7), PH patients (n=6) and control subjects (n=5).

b. Differential gene expression still lists log2FC of 0 – which is most likely the reason for identification of Tcell genes CD3, CD28, PRF1 as being differentially expressed between monocyte subsets.

Response: We apologize for the improper statement. We have revised the description as below and the revised manuscript where we wrote: “The “FindAllMarkers” function in Seurat was used to find the markers for each of the identified clusters. For each cluster, we performed functional enrichment analysis, which was implemented by clusterProfiler (v3.10.1) using the top 100 DEGs sorted by logFC in ascending order, as well as $p_{adj} \leq 0.05$.”

Identification of T cell genes *CD3*, *CD28* and *PRF1* ect. was based on an unbiased screening process in our study. First, we ranked the contributions of genes involved in each biological pathway in Figure 4d, and selected 3 genes with the largest fold change in each

biological pathway. Since different pathways might share the same genes, a total of 15 candidate genes (*AIF1*, *FCER1G*, *TGM2*, *PRF1*, *SPON2*, *TBX21*, *CLEC1B*, *NFE2*, *CLEC4A2*, *TLR2*, *CD28*, *CD3E*, *LAG3*, *GP6*, *VWF*) were determined from the 7 pathways. Second, PCR was used to determine the expression of these 15 genes in the lung tissue of hypoxic mice, the expression of 5 genes (*Prf1*, *Spon2*, *Tbx21*, *Cd28* and *Cd3e*) were significantly increased compared to that in control mice, which were consistent with the results of ScRNA sequencing in HAPH patients. At last, the 5 genes were quantitatively analyzed at the protein level, and Cd28 (CD28), Tbx21 (TBX21), and Spon2 (SPON2) were significantly elevated in the lung tissues of hypoxic PH mice, as well as in RAW264.7 monocytes/macrophages and THP-1 monocytes. This is a verification process from transcriptional level to protein level, covering all relevant pathways based on an unbiased process of screening. Therefore, the candidate genes for validation and conclusion reached are convincing, and there is no causal relationship between “differential gene expression still lists log2FC of 0” and “identification of T cell genes”.

c. The DEG should be divided into up and down-regulated genes and enriched separately.

Response: According to the valuable suggestion, we have supplemented the analysis of enrichment functions using down-regulated genes (bottom 100 of DEG sorted by logFC in ascending order, as well as $p_{adj} \leq 0.05$), as shown in **Figure R3** or Figure S17.

Figure R3. (a-c) Gene Ontology analysis (biological process) of the down-regulated genes in C0 (a), C1 (b) and C2(c).

d. The authors do not carry out any DEG analysis in other subsets. A lack of difference in frequency does not equate a lack of transcriptional changes. This is further highlighted by the follow up experiments that utilized a T cell line and endothelial cells for follow up experiments

Response: Thank you very much for the suggestions. Accordingly, we have added DEG analysis and functional enrichment in other subsets, except for monocytes, as shown in **Figure R4** or Figure S18.

(a) CD 4+ T cells

(b) CD 8+ T cells

(c) Treas

(d) NK cells

Figure R4. (a-h) Gene Ontology analysis (biological process) of the up-regulated (left) and down-regulated (right) genes in each cell type in comparison of HAPH with control group.

e. No functional assays were carried out to interrogate the differences in module scores and transcriptional profile.

Response: We agree with your professional suggestion. However, further functional assays cannot be carried out because the original ethical approval document didn't enroll this part of the new patients and Covid-19 epidemic had prevented us from reenrolling patients to carry out experiments within 3 months. Validation experiments in PAH animal models and two mononuclear/macrophage lines were conducted and confirmed the role of these genes in the process of PAH. We believe that functional validation at the cellular and animal levels can reflect the in-silico findings observed in patient blood samples.

2. Validation

a. The experiments in mouse model use total lung tissue and do not specifically target monocytes/macrophages.

Response:

Thanks for pointing out this flaw in the validation experiment. Since the mice have all been sacrificed, it is difficult for us to obtain fresh mouse blood to perform flow cytometry for sorted monocytes/macrophages due to ethical issue and COVID-19 pandemic. Instead, we performed functional validation experiments using mouse RAW264.7 monocytes/macrophages and human THP-1 monocytes. Using the hypoxia mimetic agent CoCl_2 , we established the cell model and evaluated the expression changes of Cd28 (CD28), Tbx21 (TBX21), and Spon2 (SPON2) under the condition of hypoxia. Consistent with the results from total lung tissues, Cd28 (CD28), Tbx21 (TBX21), and Spon2 (SPON2) expression levels were significantly increased in response to chemically hypoxic condition, confirming that these changed genes in monocytes/macrophages was consistent with the findings derived from scRNA-seq of human samples. Please check out the results below (Figure R5), and relevant results and original data have been added to the revised manuscript (Figure 5j-i) and supplementary materials (Figure S10).

Figure R5. (a) Immunoblottings for the expression of Cd28, Tbx21 and Spon2 in RAW264.7 monocytes/macrophages induced by CoCl_2 (150 $\mu\text{mol/L}$) or vehicle for 24 h. (b) Immunoblottings for the expression of CD28, TBX21 and SPON2 in THP-1 monocytes induced by CoCl_2 (150 $\mu\text{mol/L}$) or vehicle for 24 h. (c) Quantification of the expression of Cd28 (CD28), Tbx21 (TBX21) and Spon2 (SPON2) in RAW264.7 and THP-1 cells ($n = 4$ for per group, $*p < 0.05$, $**p < 0.01$; Student's t -test. Data were presented as mean \pm SD).

b. There is no validation for the reduced module score for HIF-1a signaling. There are antibodies that can be used to examine HIF-1a protein expression and monocyte/macrophage cell lines that can be used if primary cells are not an option.

Response:

Thanks for providing constructive comments.

We detected the HIF-1 α protein expression in lung tissues of the hypoxia-induced mice and the monocyte/macrophage cell lines induced by hypoxia-mimetic agent. Hif-1 α protein expression was significantly increased in lung tissues of chronic hypoxia-induced mice, compared to the mice under normoxia condition. Consistent with the results in animals, Hif-1 α (HIF-1 α) protein expression was also significantly increased in RAW264.7 monocytes/macrophages and THP-1 monocytes induced by hypoxia-mimetic CoCl₂, respectively. Please check out the results below (**Figure R6**), and relevant results and original data have been added to the supplementary documents (**Figure S19, S20**).

Figure R6. Hif-1 α (HIF-1 α) expression in lung tissues of the mice model and the cellular model of hypoxia-mimetic agent. (a, b) Images of immunoblottings and quantification of the expression of Hif-1 α in lung tissues of the control mice and hypoxic PH mice (n = 4 for per group, * p < 0.05; Student's t -test. Data were presented as mean \pm SD). (c, d) Images of immunoblottings and quantification of the expression of Hif-1 α in RAW264.7 monocytes/macrophages induced by CoCl₂ (150 μ mol/L) or vehicle for 24 h (n = 4 for per group, *** p < 0.001; Student's t -test. Data were presented as mean \pm SD). (e, f) Images of immunoblottings and quantification of the expression of Hif-1 α in THP-1

monocytes induced by CoCl₂ (150 μmol/L) or vehicle for 24 h (n = 4 for per group, ***p < 0.001; Student's t-test. Data were presented as mean ± SD).

Frankly, the validation results presented were not accordant with the pattern of the downregulation of HIF-1α expression in the monocytes between HAPH patients and controls in our work. The factors that lead to such inaccordance come from unique long-term high altitude anoxic environment and more significantly from different tolerance to hypoxia during altitude acclimatization. In other words, the control of high-altitude PH patients in our study was healthy people who living in high-altitude areas (hypoxia without PH), while the control of experimental hypoxic mice or hypoxic cells was under normoxia condition.

3. Data visualization minor concerns:

a. bubble plots and violin plots of marker genes are redundant and only one should be shown.

Response: We have reorganized Figure 2, Figure 3 and Figure 7, and revised the description as suggested in the revised version (revised Figure 2,3 and 7), as shown in Figure R7, R8, R9.

Figure R7. Differential immune cell composition in the circulation of patients with HAPH. (a) t-SNE (t-distributed stochastic neighbor embedding) plot of the main immune cell subsets. (b) Dot plot depicting the percentages and average expressions of the canonical genes associated with each main immune cell cluster. (c) Boxplots comparing the percentages of indicated cell types in PBMCs between HAPH patients (n=7) and control subjects (n=5). The two-sided p values from the Wilcoxon rank-sum test were shown. (d). Proportions of each cell type in HAPH patients (n=7) and control subjects (n=5).

Figure R8. Immune cell composition in the circulation system of patients with PH and its comparison with those in HAPH. (a) t-SNE plot of the main immune cell subsets in PBMCs from patients with PH (n=6). (b) Dot plot depicting the percentages and average expressions of the canonical genes associated with each main immune cell cluster in PBMCs from patients with PH (n=6). (c) Boxplots comparing the percentages of each main cell type in PBMCs between HAPH (n=7) and PH (n=6) patients. The two-sided p values from the Wilcoxon rank-sum test were shown. (d) Proportions of each cell type in each sample as indicated. (e-g) The proportion of T cells, NK cells and C0, C1, C2 monocyte subsets in PBMCs from HAPH (n=11) and PH (n=6) patients as analyzed by flow cytometry (*p < 0.05; Unpaired t test was utilized as appropriate. Data were presented as mean ± SD).

Figure R9 (Figure 3). Expansion of monocyte clusters in circulation system of patients with HAPH. (a) t-SNE plot of all monocytes collected in the present study. (b) t-SNE plot of all monocytes and the sub-clusters with cells colored based upon canonical gene markers of monocytes. clusters 0 (C0) represented as classical monocytes, C1 as non-classical monocytes and C2 as intermediate monocytes. (c) Heatmap displaying the scaled expression values of discriminative gene sets from all monocytes in PBMCs between HAPH and control subjects. The top 50 marker genes in each subgroup were shown. (d) Dot plot depicting the percentages and average expressions of the classical genes associated with each monocyte cluster. (e) Proportions of each monocyte cluster in each sample as indicated. (f) Boxplots comparing the percentages of each monocyte cluster in PBMCs between HAPH patients (n=7) and control subjects (n=5). The two-sided p values

from the Wilcoxon rank-sum test were shown. (g) Gating strategy of Monocytes by flow cytometry. (h) The comparison of the proportion of each monocyte subsets in PBMCs between HAPH patients (n=11) and controls (n=10) assessed by flow cytometry. (*p < 0.05; Unpaired t test was utilized as appropriate. Data were presented as mean ± SD).

b. Individual samples should be put in the supplemental figures.

Response: We have supplemented **Figure R10** or Figure S16 in the revised supplementary materials.

(a)

(b)

Figure R10. tSNE plots of individual samples in HAPH (H) and Control (C) group (a) and in HAPH (H) and PH (P) group (b).

c. Feature plots should also be provided as supplemental figures rather than main figure – rather a main UMAP with clusters identified should be provided.

Response: We have reorganized Figure 2 as suggested in the revised version (Figure R11 or revised Figure 2).

Figure R11. Differential immune cell composition in the circulation of patients with HAPH. (a) t-SNE (t-distributed stochastic neighbor embedding) plot of the main immune cell subsets. (b) Dot plot depicting the percentages and average expressions of the canonical genes associated with each main immune cell cluster. (c) Boxplots comparing the percentages of indicated cell types in PBMCs between HAPH patients (n=7) and control subjects (n=5). The two-sided p values from the Wilcoxon rank-sum test were shown. (d). Proportions of each cell type in HAPH patients (n=7) and control subjects (n=5).

d. Heat map of marker genes for monocyte subsets without any gene names is not informative – there is no reason to show figure 2C- this more complete list can be provided as a supplementary table.

Response: We have provided the complete gene list as supplementary materials (Table S10) as suggested.

Table S10. The differential expression gene list of three Monocyte subcluster C0,C1,C2.

Monocyte_C0	Monocyte_C1	Monocyte_C2
S100A9	PPBP	IL32
CD14	CD68	NKG7
S100A8	FCGR3A	GNLY
LYZ	MS4A7	CTSW
S100A12	LST1	CST7
VCAN	MYL9	CD3E
MS4A6A	IFITM3	IFITM1

CSTA	RHOC	GZMH
MNDA	TIMP1	GZMB
FCN1	TPM1	PRF1
NFKBIA	PECAM1	GZMA
NCF1	TREML1	CD69
FOS	SMIM25	CCL5
TYMP	TPM4	FGFBP2
LGALS2	HIST1H2AC	PCED1B-AS1
CEBPD	FAM110A	CD7
BLVRB	ITGA2B	TRBC2
PLBD1	SERPINA1	CD247
MAFB	WARS	GZMM
VIM	C19orf33	KLRB1

e. Gating strategy for monocytes is highly unusual – it should be a CD14 versus CD16 dot plot

Response: Thanks to the reviewer for helpful comments. At the beginning of our study, we also considered to use of CD14 versus CD16 as a gating strategy for monocytes on this subject. However, we found that this strategy was not achievable.

Firstly, the classification of Monocyte subpopulation markers shows that strong positive CD14 represents Monocyte_C0, CD16/FCGR3A positive represents Monocyte_C1 and CD14 negative and CD16 negative or weak positive represents Monocyte_C2, as shown in **Figure R12d** or Figure 3d. If only the classical CD14 and CD16 markers are used as the gate strategy for total Monocytes, C0 and C1 subgroup can be separated, but the C2 subgroup cannot be distinguished.

Secondly, according to the result of tSNE, CD68 can be used as a marker of total Monocytes in this project (**Figure R13**), and there are also reports suggesting that CD68 can be used as a marker of total Monocytes (PMID: 32783921; PMID: 32783921; PMCID: PMC7368915).

Therefore, in order to corroborate the results of single-cell RNA sequencing data in this study and distinguish different subsets of Monocyte, we did not use CD14 and CD16, but use CD68 as total monocytes instead. Leukocyte subsets were analyzed using the following combination of surface markers: classical monocytes (C0) as CD68+CD14+, non-classical monocytes (C1) as CD68+CD14-CD16+ and intermediate monocytes (C2) as CD68+CD14-CD16- (**Figure R12g** or Figure 3g).

Figure R12 (Figure 3). Expansion of monocyte clusters in circulation system of patients with HAPH. (a) t-SNE plot of all monocytes collected in the present study. (b) t-SNE plot of all monocytes and the sub-clusters with cells colored based upon canonical gene markers of monocytes. clusters 0 (C0) represented as classical monocytes, C1 as non-classical monocytes and C2 as intermediate monocytes. (c) Heatmap displaying the scaled expression values of discriminative gene sets from all monocytes in PBMCs between HAPH and control subjects. The top 50 marker genes in each subgroup were shown. (d) Dot plot depicting the percentages and average expressions of the classical genes associated with each monocyte cluster. (e) Proportions of each monocyte cluster in each sample as indicated. (f) Boxplots comparing the percentages of each monocyte cluster in

PBMCs between HAPH patients (n=7) and control subjects (n=5). The two-sided p values from the Wilcoxon rank-sum test were shown. (g) Gating strategy of Monocytes by flow cytometry. (h) The comparison of the proportion of each monocyte subsets in PBMCs between HAPH patients (n=11) and controls (n=10) assessed by flow cytometry. (*p < 0.05; Unpaired t test was utilized as appropriate. Data were presented as mean \pm SD).

Figure R13. The feature plot of CD68 in Monocytes.

REVIEWERS' COMMENTS

Reviewer #3 (Remarks to the Author):

the authors have now addressed the concerns that persisted in the first revision. i have no further concerns.